# Revisiting the Message Passing in Heterophilous Graph Neural Networks

## Abstract

Graph Neural Networks (GNNs) have demonstrated strong performance in graph mining tasks due to their message-passing mechanism, which is aligned with the homophily assumption that adjacent nodes exhibit similar behaviors. However, in many real-world graphs, connected nodes may display contrasting behaviors, termed as *heterophilous* patterns, which has attracted increased interest in heterophilous GNNs (HTGNNs). Although the message-passing mechanism seems unsuitable for heterophilous graphs due to the propagation of class-irrelevant information, it is still widely used in many existing HTGNNs and consistently achieves notable success. This raises the question: *why does message passing remain effective on heterophilous graphs?* To answer this question, in this paper, we revisit the message-passing mechanisms in heterophilous graph neural networks and reformulate them into a unified heterophilous message-passing (HTMP) mechanism. Based on HTMP and empirical analysis, we reveal that the success of message passing in existing HTGNNs is attributed to implicitly enhancing the compatibility matrix among classes. Moreover, we argue that the full potential of the compatibility matrix is not completely achieved due to the existence of incomplete and noisy semantic neighborhoods in real-world heterophilous graphs. To bridge this gap, we introduce a new approach named CMGNN, which operates within the HTMP mechanism to explicitly leverage and improve the compatibility matrix. A thorough evaluation involving 10 benchmark datasets and comparative analysis against 13 well-established baselines highlights the superior performance of the HTMP mechanism and CMGNN method.

## 1 Introduction

Graph Neural Networks (GNNs) have shown remarkable performance in graph mining tasks, such as social network analysis [1, 2] and recommender systems [3, 4]. The design principle of GNNs is typically based on the homophily assumption [5], which assumes that nodes are inclined to exhibit behaviors similar to their neighboring nodes [6]. However, this assumption does not always hold in real-world graphs, where the connected nodes demonstrate a contrasting tendency known as the *heterophily* [7]. In response to the challenges of heterophily in graphs, *heterophilous GNNs (HTGNNs)* have attracted considerable research interest [6, 8–10], with numerous innovative approaches being introduced recently [11–24]. However, the majority of these methods continue to employ a message-passing mechanism, which was not originally designed for heterophilous graphs, as they tend to incorporate excessive information from disparate classes. This naturally raises a question: *Why does message passing remain effective on heterophilous graphs?*

Recently, a few efforts [6] have begun to investigate this question and reveal that vanilla message passing can work on heterophilous graphs under certain conditions. However, the absence of a unified

and comprehensive understanding of message passing within existing HTGNNs has hindered the creation of innovative approaches. In this paper, we first revisit the message-passing mechanisms in existing HTGNNs and reformulate them into a unified heterophilous message-passing (HTMP) mechanism, which extends the definition of neighborhood in various ways and simultaneously utilizes the messages of multiple neighborhoods. Specifically, HTMP consists of three major steps namely aggregating messages with explicit guidance, combining messages from multiple neighborhoods, and fusing intermediate representations.

Equipped with HTMP, we further conduct empirical analysis on real-world graphs. The results reveal that the success of message passing in existing HTGNNs is attributed to **implicitly enhancing the compatibility matrix**, which exhibits the probabilities of observing edges among nodes from different classes. In particular, by increasing the distinctiveness between the rows of the compatibility matrix via different strategies, the node representations of different classes become more discriminative in heterophilous graphs.

Drawing from previous observations, we contend that nodes within real-world graphs might exhibit a semantic neighborhood that only reveals a fraction of the compatibility matrix, accompanied by noise. This could limit the effectiveness of enhancing the compatibility matrix and result in suboptimal representations. To fill this gap, we further propose a novel Compatibility Matrix-aware Graph Neural Network (CMGNN) under HTMP mechanism, which utilizes the compatibility matrix to construct desired neighborhood messages as supplementary for nodes and explicitly enhances the compatibility matrix by a targeted constraint. We build a benchmark to fairly evaluate CMGNN and existing methods, which encompasses 13 diverse baseline methods and 10 datasets that exhibit varying levels of heterophily. Extensive experimental results demonstrate the superiority of CMGNN and HTMP mechanism. The contributions of this paper are summarized as:

- We revisit the message-passing mechanisms in existing HTGNNs and reformulate them into a unified heterophilous message-passing mechanism (HTMP), which not only provides a macroscopic view of message passing in HTGNNs but also enables people to develop new methods flexibly.

- We reveal that the effectiveness of message passing on heterophilous graphs is attributed to implicitly enhancing the compatibility matrix among classes, which gives us a new perspective to understand the message passing in HTGNNs.

- Based on HTMP mechanism and empirical analysis, we propose CMGNN to unlock the potential of the compatibility matrix in HTGNNs. We further build a unified benchmark that overcomes the issues of current datasets for fair evaluation[1]. Experiments show the superiority of CMGNN.

## 2 Preliminaries

Given a graph $\mathcal{G} = (\mathcal{V}, \mathcal{E}, \mathbf{X}, \mathbf{A}, \mathbf{Y})$, $\mathcal{V}$ is the node set and $\mathcal{E}$ is the edge set. Nodes are characterized by the feature matrix $\mathbf{X} \in \mathbb{R}^{N \times d_f}$, where $N = |\mathcal{V}|$ denotes the number of nodes, $d_f$ is the features dimension. $\mathbf{Y} \in \mathbb{R}^{N \times 1}$ is the node labels with the one-hot version $\mathbf{C} \in \mathbb{R}^{N \times K}$, where $K$ is the number of node classes. The neighborhood of node $v_i$ is denoted as $\mathcal{N}_i$. $\mathbf{A} \in \mathbb{R}^{N \times N}$ is the adjacency matrix , and $\mathbf{D} = \text{diag}(\mathbf{d}_1, ..., \mathbf{d}_n)$ represents the diagonal degree matrix, where $\mathbf{d}_i = \sum_j \mathbf{A}_{ij}$. $\tilde{\mathbf{A}} = \mathbf{A} + \mathbf{I}$ represents the adjacency matrix with self-loops. Let $\mathbf{Z} \in \mathbb{R}^{N \times d_r}$ be the node representations with dimension $d_r$ learned by the models. We use $\mathbf{1}$ to represent a matrix with all elements equal to 1, and $\mathbf{0}$ for a matrix with all elements equal to 0.

**Homophily and Heterophily**. High homophily is observed in graphs where a substantial portion of connected nodes shares identical labels, while high heterophily corresponds to the opposite situation. For measuring the homophily level, two widely used metrics are edge homophily $h^e$ [12] and node homophily $h^n$ [15], defined as $h^e = \frac{|\{e_{u,v}|e_{u,v} \in \mathcal{E}, \mathbf{Y}_u = \mathbf{Y}_v\}|}{|\mathcal{E}|}$ and $h^n = \frac{1}{|\mathcal{V}|} \sum_{v \in \mathcal{V}} \frac{|\{u|u \in \mathcal{N}_v, \mathbf{Y}_u = \mathbf{Y}_v\}|}{\mathbf{d}_v}$. Both metrics have a range of $[0, 1]$, where higher values indicate stronger homophily and lower values indicate stronger heterophily.

**Vanilla Message Passing (VMP)**. The vanilla message-passing mechanism plays a pivotal role in transforming and updating node representations based on the neighborhood [25]. Typically, the

---

[1]Codebase is available at the supplementary material.

Table 1: Revisiting the message passing in representative heterophilous GNNs under the perspective of HTMP mechanism.

| Method | Neighborhood Indicators | | Aggregation Guidance | | COMBINE | FUSE |
|---|---|---|---|---|---|---|
| | Type | $\mathcal{A}$ | Type | $\mathcal{B}$ | | |
| GCN [1] | Raw | $[\tilde{\mathbf{A}}]$ | DegAvg | $[\tilde{\mathbf{B}}^d]$ | / | $\mathbf{Z} = \mathbf{Z}^L$ |
| APPNP [26] | | $[\mathbf{I}, \tilde{\mathbf{A}}]$ | | $[\mathbf{I}, \tilde{\mathbf{B}}^d]$ | WeightedAdd | $\mathbf{Z} = \mathbf{Z}^L$ |
| GCNII [27] | | $[\mathbf{I}, \tilde{\mathbf{A}}]$ | | $[\mathbf{I}, \tilde{\mathbf{B}}^d]$ | WeightedAdd | $\mathbf{Z} = \mathbf{Z}^L$ |
| GAT [28] | | $[\tilde{\mathbf{A}}]$ | AdaWeight | $[\mathbf{B}^{aw}]$ | / | $\mathbf{Z} = \mathbf{Z}^L$ |
| GPR-GCN [20] | Raw | $[\tilde{\mathbf{A}}]$ | | $[\tilde{\mathbf{B}}^d]$ | / | AdaAdd |
| OrderedGNN [21] | | $[\mathbf{I}, \mathbf{A}]$ | DegAvg | $[\mathbf{I}, \mathbf{B}^d]$ | AdaCat | $\mathbf{Z} = \mathbf{Z}^L$ |
| ACM-GCN [18] | | $[\mathbf{I}, \mathbf{A}, \tilde{\mathbf{A}}]$ | | $[\mathbf{I}, \mathbf{B}^d, \mathbf{I} - \mathbf{B}^d]$ | AdaAdd | $\mathbf{Z} = \mathbf{Z}^L$ |
| FAGCN [11] | | $[\mathbf{I}, \mathbf{A}]$ | AdaWeight | $[\mathbf{I}, \mathbf{B}^{naw}]$ | WeightedAdd | $\mathbf{Z} = \mathbf{Z}^L \mathbf{W}$ |
| GBK-GNN [24] | | $[\mathbf{I}, \mathbf{A}, \mathbf{A}]$ | | $[\mathbf{I}, \mathbf{B}^{aw}, \mathbf{1} - \mathbf{B}^{aw}]$ | Add | $\mathbf{Z} = \mathbf{Z}^L$ |
| SimP-GCN [14] | ReDef | $[\mathbf{I}, \tilde{\mathbf{A}}, \mathbf{A}_f]$ | | $[\mathbf{I}, \tilde{\mathbf{B}}^d, \mathbf{B}_f^d]$ | AdaAdd | $\mathbf{Z} = \mathbf{Z}^L$ |
| H2GCN [12] | | $[\mathbf{A}, \mathbf{A}_{h2}]$ | DegAvg | $[\mathbf{B}^d, \mathbf{B}_{h2}^d]$ | Cat | Cat |
| Geom-GCN [15] | | $[\mathbf{A}_{c1}, ..., \mathbf{A}_{cr}, ..., \mathbf{A}_{cR}]$ | | $[\mathbf{B}_{c1}^d, ..., \mathbf{B}_{cr}^d, ..., \mathbf{B}_{cR}^d]$ | Cat | $\mathbf{Z} = \mathbf{Z}^L$ |
| MixHop [16] | | $[\mathbf{I}, \mathbf{A}, \mathbf{A}_{h2}, ..., \mathbf{A}_{hk}]$ | | $[\mathbf{I}, \mathbf{B}^d, \mathbf{B}_{h2}^d, ..., \mathbf{B}_{hk}^d]$ | Cat | $\mathbf{Z} = \mathbf{Z}^L$ |
| UGCN [13] | ReDef | $[\tilde{\mathbf{A}}, \tilde{\mathbf{A}}_{h2}, \mathbf{A}_f]$ | AdaWeight | $[\tilde{\mathbf{B}}^{aw}, \tilde{\mathbf{B}}_{h2}^{aw}, \mathbf{B}_f^{aw}]$ | AdaAdd | $\mathbf{Z} = \mathbf{Z}^L$ |
| WRGNN [22] | | $[\mathbf{A}_{c1}, ..., \mathbf{A}_{cr}, ..., \mathbf{A}_{cR}]$ | | $[\mathbf{B}_{c1}^{aw}, ..., \mathbf{B}_{cr}^{aw}, ..., \mathbf{B}_{cR}^{aw}]$ | Add | $\mathbf{Z} = \mathbf{Z}^L$ |
| HOG-GCN [17] | | $[\mathbf{I}, \mathbf{A}_{hk}]$ | | $[\mathbf{I}, \mathbf{B}^{re}]$ | WeightedAdd | $\mathbf{Z} = \mathbf{Z}^L$ |
| GloGNN [19] | | $[\mathbf{I}, \mathbf{1}]$ | RelaEst | $[\mathbf{I}, \mathbf{B}^{re}]$ | WeightedAdd | $\mathbf{Z} = \mathbf{Z}^L$ |
| GGCN [23] | Dis | $[\mathbf{I}, \mathbf{A}_p, \mathbf{A}_n]$ | | $[\mathbf{I}, \mathbf{B}_p^{re}, \mathbf{B}_n^{re}]$ | AdaAdd | $\mathbf{Z} = \mathbf{Z}^L$ |

*   The correspondence between the full form and the abbreviation: Raw Neighborhood (Raw), Neighborhood Redefine (ReDef), Neighborhood Discrimination (Dis), Degree-based Averaging (DegAvg), Adaptive Weights (AdaWeight), Relation Estimation (RelaEst), Addition (Add), Weighted Addition (WeightAdd), Adaptive Weighted Addition (AdaAdd), Concatenation (Cat), Adaptive Dimension Concatenation (AdaCat).
*   More details about the notations are available in Appendix A.1.

mechanism operates iteratively and comprises two stages:

$$\widetilde{\mathbf{Z}}^l = \text{AGGREGATE}(\mathbf{A}, \mathbf{Z}^{l-1}), \quad \mathbf{Z}^l = \text{COMBINE}\left(\mathbf{Z}^{l-1}, \widetilde{\mathbf{Z}}^l\right), \tag{1}$$

where the AGGREGATE function first aggregates the input messages $\mathbf{Z}^{l-1}$ from neighborhood $\mathbf{A}$ into the aggregated one $\widetilde{\mathbf{Z}}^l$, and subsequently, the COMBINE function combines the messages of node ego and neighborhood aggregation, resulting in updated representations $\mathbf{Z}^l$.

## 3 Revisiting Message Passing in Heterophilous GNNs.

To gain a thorough and unified insight into the effectiveness of message passing in HTGNNs, we revisit message passing in various notable HTGNNs [11–24] and propose a unified heterophilous message passing (HTMP) mechanism, structured as follows:

$$\widetilde{\mathbf{Z}}_r^l = \text{AGGREGATE}(\mathbf{A}_r, \mathbf{B}_r, \mathbf{Z}^{l-1}), \ \mathbf{Z}^l = \text{COMBINE}(\{\widetilde{\mathbf{Z}}_r^l\}_{r=1}^R), \ \mathbf{Z} = \text{FUSE}(\{\mathbf{Z}^l\}_{l=0}^L). \tag{2}$$

Generally, HTMP extends the definition of neighborhood in various ways and simultaneously utilize the messages of multiple neighborhoods, which is the key for better adapting to heterophily. We use $R$ to denote the number of neighborhoods used by the model. In each message passing layer $l$, HTMP separately aggregates messages within $R$ neighborhoods and combines them. The methodological analysis of some representative HTGNNs and more details can be seen in Appendix A. Compared to the VMP mechanism, HTMP mechanism has advances in the following functions:

(i) To characterize different neigborhoods, the **AGGREGATE** function in HTMP includes the **neighborhood indicator** $\mathbf{A}_r$ to indicate the neighbors within a specific neighborhood $r$. The adjacency matrix $\mathbf{A}$ in VMP is a special neighborhood indicator that marks the neighbors in the raw neighborhood. To further characterize the aggregation of different neighborhoods, HTMP introduces the **aggregation guidance** $\mathbf{B}_r$ for each neighborhood $r$. In VMP, the aggregation guidance is an implicit parameter of the AGGREGATE function since it only works for the raw neighborhood. A commonly used form of the AGGREGATE function is $\text{AGGREGATE}(\mathbf{A}_r, \mathbf{B}_r, \mathbf{Z}^{l-1}) = (\mathbf{A}_r \odot \mathbf{B}_r)\mathbf{Z}^{l-1}\mathbf{W}_r^l$, where $\odot$ is the Hadamard product and $\mathbf{W}_r^l$ is a weight matrix for message transformation. We take

this as the general form of the AGGREGATE function and only analyze the neighborhood indicators and the aggregation guidance in the following.

The *neighborhood indicator* $\mathbf{A}_r \in \{0,1\}^{N \times N}$ indicates neighbors associated with central nodes within neighborhood $r$. To describe the multiple neighborhoods in HTGNNs, neighborhood indicators can be formed as a list $\mathcal{A} = [\mathbf{A}_1, ..., \mathbf{A}_r, ..., \mathbf{A}_R]$. For the sake of simplicity, we consider the identity matrix $\mathbf{I} \in \mathbb{R}^{N \times N}$ as a special neighborhood indicator for acquiring the nodes' ego messages. The *aggregation guidance* $\mathbf{B}_r \in \mathbb{R}^{N \times N}$ can be viewed as pairwise aggregation weights in most cases, which has the multiple form $\mathcal{B} = [\mathbf{B}_1, ..., \mathbf{B}_r, ..., \mathbf{B}_R]$. Table 1 illustrates the connection between message passing in various HTGNNs and HTMP mechanism.

(ii) Considering the existence of multiple neighborhoods, the **COMBINE** function in HTMP need to integrate multiple messages instead of only the ego node and the raw neighborhood. Thus, the input of the COMBINE function is a set of messages $\widetilde{\mathbf{Z}}_r^l$ aggregated from the corresponding neighborhoods. In HTGNNs, addition and concatenation are two common approaches, each of which has variants. An effective COMBINE function is capable of simultaneously processing messages from various neighborhoods while preserving their distinct features, thereby reducing the effects of heterophily.

(iii) In VMP, the final output representations are usually the one of the final layer: $\mathbf{Z} = \mathbf{Z}^L$. Some HTGNNs utilize the combination of intermediate representations to leverage messages from different localities, adapting to the heterophilous structural properties in different graphs. Thus, we introduce an additional **FUSE** function in HTMP which integrates multiple representations $\mathbf{Z}^l$ of different layers $l$ into the final $\mathbf{Z}$. Similarly, the FUSE function is based on addition and concatenation.

# 4 Why Does Message Passing Still Remain Effective in Heterophilous Graphs?

Based on HTMP mechanism, we further dive into the motivation behind the message passing of existing HTGNNs. Our discussion begins by examining the difference between homophilous and heterophilous graphs. Initially, we consider the homophily ratios $h^e$ and $h^n$, as outlined in Section 2. However, a single number is not able to indicate enough conditions of a graph. Ma et al. [6] propose the existence of a special case of heterophily, named "good" heterophily, where the VMP mechanism can achieve strong performance and the homophily ratio shows no difference. Thus, to better study the heterophily property, here we introduce the *Compatibility Matrix* [7] to describe graphs:

**Definition 1** *Compatibility Matrix (CM): The potential connection preference among classes within a graph. It's formatted as a matrix $\mathbf{M} \in \mathbb{R}^{K \times K}$, where the $i$-th row $\mathbf{M}_i$ denotes the connection probabilities between class $i$ and all classes. It can be estimated empirically by the statistics among nodes as follows:*

$$\mathbf{M} = Norm(\mathbf{C}^T \mathbf{C}^{nb}), \quad \mathbf{C}^{nb} = \hat{\mathbf{A}}\mathbf{C}, \tag{3}$$

*where $Norm(\cdot)$ denotes the L1 normalization and $T$ is the matrix transpose operation. $\mathbf{C}^{nb} \in \mathbf{R}^{N \times K}$ is the **semantic neighborhoods** of nodes, which indicates the proportion of neighbors from each class in nodes' neighborhoods.*

We visualize the CM of a homophilous graph Photo [29] and a heterophilous graph Amazon-Ratings [30] in Figure 1(a) and 1(b). The CM in Photo displays an identity-like matrix, where the diagonal elements can be viewed as the homophily level of each class. With this type of CM, the VMP mechanism learns representations comprised mostly of messages from same the class, while messages of other classes are diluted. *Then how does HTMP mechanism work on heterophilous graphs without an identity-like CM?* The "good" heterophily inspires us, which we believe corresponds to a CM with enough discriminability among classes. We conduct experiments on synthetic graphs to confirm this idea, with details available in Appendix C. Also, we find "good" heterophily in real-world graphs though it's not as significant as imagined. Thus, we have the following observation:

**Observation 1** *(Connection between CM and VMP). When enough (depends on data) discriminability exists among classes in CM, vanilla message passing can work well in heterophilous graphs.*

With this observation, we have a conjecture: *Is HTMP mechanism trying to enhance the discriminability of CM?* Some special designs in HTMP intuitively meet this. For example, *feature-similarity-based neighborhood indicators* and *neighborhood discrimination* are designed to construct neighborhoods

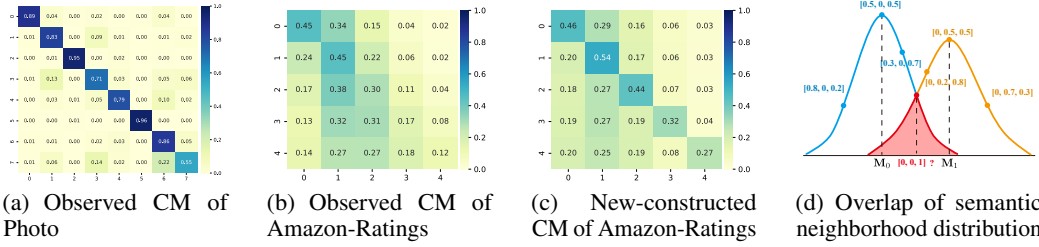

(a) Observed CM of Photo

(b) Observed CM of Amazon-Ratings

(c) New-constructed CM of Amazon-Ratings

(d) Overlap of semantic neighborhood distribution

Figure 1: Visualizations of the compatibility matrix and the example of distribution overlap.

with high homophily, that is, an identity-like CM with high discriminability. We plot the CM of feature-similarity-based neighborhood on Amazon-Ratings in Figure 1(c) to confirm it. Moreover, we investgate two representative methods ACM-GCN [18] and GPRGNN [20], showing that they also meet this conjecture with the posterior proof in Appendix D. ACM-GCN combines the messages of node ego, low-frequency and high-frequency with adaptive weights, which actually motifs the edge weights and node weights to build a new CM. GPRGNN has a FUSE function with adaptive weights while other settings are the same as GCN. It actually integrates the CMs of multiple-order neighborhoods with adaptive weights to form a more discriminative CM. These lead to the answer to the aforementioned question:

**Observation 2** *(Connection between CM and HTMP). The unified goal of various message passing in existing HTGNNs is to utilize and enhance the discriminability of CM on heterophilous graphs. In other words, the success of message passing in existing HTGNNs benefits from utilizing and enhancing the discriminability of CM.*

Furthermore, we notice that the power of CM is not fully released due to the incomplete and noisy semantic neighborhoods in real-world heterophilous graphs. We use the perspective of distribution to describe the issue more intuitively: The semantic neighborhoods of nodes from the same class collectively form a distribution, whose mean value indicates the connection preference of that class, i.e. $\mathbf{M}_i$ for class $i$. Influenced by factors such as degree and randomness, the semantic neighborhood of nodes in real-world graphs may display only a fraction of CM accompanied by noise. It can lead to the overlap between different distributions as shown in Figure 1(d), where the existence of overlapping parts means nodes from different classes may have the same semantic neighborhood. This brings a great challenge since the overlapping semantic neighborhood may become redundant information during message passing.

# 5   Method

To fill this gap, we further propose a method named Compatibility Matrix-Aware GNN (CMGNN), which leverages the CM to construct desired neighborhood messages as supplementary, providing valuable neighborhood information for nodes to mitigate the impact of incomplete and noisy semantic neighborhoods. The desired neighborhood message denotes the averaging message within a neighborhood when a node's semantic neighborhoods meet the CM of the corresponding class, which converts the discriminability from CM into messages. CMGNN follows the HTMP mechanism and constructs a supplementary neighborhood indicator along with the corresponding aggregation guidance to introduce supplementary messages. Further, CMGNN introduces a simple constraint to explicitly enhance the discriminability of CM.

**Message Passing in CMGNN.**   CMGNN aggregates messages from three neighborhoods for each node, including the ego neighborhood, raw neighborhood, and supplementary neighborhood. Following the HTMP mechanism, the message passing of CMGNN cen be described as follows:

$$\widetilde{\mathbf{Z}}_r^l = \text{AGGREGATE}(\mathbf{A}_r, \mathbf{B}_r, \mathbf{Z}^{l-1}) = (\mathbf{A}_r \odot \mathbf{B}_r)\mathbf{Z}^{l-1}\mathbf{W}_r^l,$$

$$\mathbf{Z}^l = \text{COMBINE}(\{\widetilde{\mathbf{Z}}_r^l\}_{r=1}^3) = \text{AdaWeight}(\{\widetilde{\mathbf{Z}}_r^l\}_{r=1}^3),$$

$$\mathbf{Z} = \text{FUSE}(\{\mathbf{Z}^l\}_{l=0}^L) = \underset{l=0}{\overset{L}{\|}} \mathbf{Z}^l,$$

(4)

where AdaWeight is the adaptive weighted addition implemented by an MLP with Softmax, $\|$ denotes the concatenation. The neighborhood indicators and aggregation guidance of the three neighborhoods are formatted as follows:

$$\mathbf{A}_1^l = \mathbf{I}, \ \mathbf{B}_1^l = \mathbf{I}, \quad \mathbf{A}_2^l = \mathbf{A}, \ \mathbf{B}_2^l = \mathbf{D}^{-1}\mathbf{1}, \quad \mathbf{A}_3^l = \mathbf{A}^{sup}, \ \mathbf{B}_3^l = \mathbf{B}^{sup}, \tag{5}$$

where $\mathbf{A}^{sup}$ and $\mathbf{B}^{sup}$ are described below.

The supplementary neighborhood indicator $\mathbf{A}^{sup}$ assigns $K$ additional virtual neighbors for each node: $\mathbf{A}^{sup} = \mathbf{1} \in \mathbb{R}^{N \times K}$. Specifically, these additional neighbors are $K$ virtual nodes, constructed as the prototypes of classes based on the labels of the training set. The attributes $\mathbf{X}^{ptt} \in \mathbb{R}^{K \times d_f}$, neighborhoods $\mathbf{A}^{ptt} \in \mathbb{R}^{K \times N}$ and labels $\mathbf{Y}^{ptt} \in \mathbb{R}^{K \times K}$ of prototypes are defined as follows:

$$\mathbf{X}^{ptt} = \mathrm{Norm}(\mathbf{C}_{train}{}^T \mathbf{X}_{train}), \ \mathbf{A}^{ptt} = \mathbf{0}, \ \mathbf{Y}^{ptt} = \mathbf{I}, \tag{6}$$

where $\mathbf{C}_{train}$ and $\mathbf{X}_{train}$ are the one-hot labels and attributes of nodes in the training set. Utilizing class prototypes as supplementary neighborhoods can provide each node with representative messages of classes, which builds the basis for desired neighborhood messages.

The supplementary aggregation guidance $\mathbf{B}^{sup} = \hat{\mathbf{C}}\hat{\mathbf{M}}$ indicates the desired semantic neighborhood of nodes, i.e. the desired proportion of neighbors from each class in nodes' neighborhoods according to the probability that nodes belong to each class. $\hat{\mathbf{M}}$ is the estimated compatibility matrix described in below. Using soft logits instead of one-hot pseudo labels preserves the real characteristics of nodes and reduces the impact of wrong predictions. During the message aggregation in the supplementary neighborhoods, the input representations $\mathbf{Z}^{l-1}$ are replaced by the representations of virtual prototype nodes $\mathbf{Z}_{ptt}^{l-1}$, which are obtained by the same message-passing mechanism as real nodes.

Similar to existing methods [18, 19], we also regard topology structure as a kind of additional available node features. Thus, the input representation of the first layer can be obtained in two ways:

$$\mathbf{Z}^0 = [\mathbf{X}\mathbf{W}^X \| \hat{\mathbf{A}}\mathbf{W}^A]\mathbf{W}^0, \ \text{or} \ \mathbf{Z}^0 = \mathbf{X}\mathbf{W}^0. \tag{7}$$

Note that in practice, we use ReLU as the activation function between layers. From the perspective of HTMP mechanism, our special design is to introduce an additional neighborhood indicator $\mathbf{A}^{sup}$ by neighborhood redefining and aggregation guidance $\mathbf{B}^{sup}$, which can be seen as a form of relation estimation along with good interpretability. Meanwhile, these designs greatly reduce the time and space cost via the $N \times K$ form.

**Compatibility Matrix Estimation.** The CM can be directly calculated via Eq 3 with full-available labels. However, the label information is not entirely available in semi-supervised settings. Thus, we try to estimate the CM with the help of semi-supervised and pseudo labels. Since the pseudo labels predicted by the model might be wrong, which can lead to low-quality estimation, we introduce the confidence $\mathbf{g} \in \mathbb{R}^{N \times 1}$ based on the information entropy to reduce the impact of wrong predictions, where a high entropy means low confidence:

$$\mathbf{g}_i = \log K - \mathrm{H}(\hat{\mathbf{C}}_i) \in [0, \log K], \tag{8}$$

where $\hat{\mathbf{C}} \in \mathbb{R}^{N \times K}$ is the soft pseudo labels composed of labels from the training set and model predictions. Then the nodes' semantic neighborhoods $\mathbf{C}^{nb} = \mathrm{Norm}(\mathbf{A}(\mathbf{g} \cdot \hat{\mathbf{C}})) \in \mathbb{R}^{N \times K}$ are calculated considering the confidence.

Further, the degrees of nodes also influence the estimation. As we mentioned in Section 4, the semantic neighborhood of low-degree nodes may display incomplete CM, leading to a significant gap between semantic neighborhoods and corresponding CM. Thus, they deserve low weights during the estimation. We manually set up two fixed thresholds and a weighting function range in $[0, 1]$:

$$\mathbf{w}_i^d = \begin{cases} \mathbf{d}_i/2K, & \mathbf{d}_i \leq K, \\ 0.25 + \mathbf{d}_i/4K, & K < \mathbf{d}_i \leq 3K, \\ 1, & otherwise. \end{cases} \tag{9}$$

When a node's degree $\mathbf{d}_i$ is smaller than the number of classes $K$, its semantic neighborhood is unlikely to display complete CM, corresponding to a low weight. And when the node degree is greater than $3K$, we believe it can display near-complete CM, corresponding to the maximum weight. Finally, we can estimate the compatibility matrix $\hat{\mathbf{M}} \in \mathbb{R}^{K \times K}$ as follows:

$$\hat{\mathbf{M}} = \mathrm{Norm}((\mathbf{w}^d \cdot \mathbf{g} \cdot \hat{\mathbf{C}})^T)\mathbf{C}^{nb}. \tag{10}$$

Table 2: Node classification accuracy comparison (%). The error bar (±) denotes the standard deviation of results over 10 trial runs. The best and second-best results in each column are highlighted in **bold** font and underlined. OOM denotes out-of-memory error during the model training.

| Dataset | Roman-Empire | Amazon-Ratings | Chameleon-F | Squirrel-F | Actor | Flickr | BlogCatalog | Wikics | Pubmed | Photo | Avg. Rank |
|---|---|---|---|---|---|---|---|---|---|---|---|
| **Homo.** | 0.05 | 0.38 | 0.25 | 0.22 | 0.22 | 0.24 | 0.4 | 0.65 | 0.8 | 0.83 | |
| **Nodes** | 22,662 | 24,492 | 890 | 2,223 | 7,600 | 7,575 | 5,196 | 11,701 | 19,717 | 7,650 | |
| **Edges** | 65,854 | 186,100 | 13,584 | 65,718 | 30,019 | 479,476 | 343,486 | 431,206 | 88,651 | 238,162 | |
| **Classes** | 18 | 5 | 5 | 5 | 5 | 9 | 6 | 10 | 3 | 8 | |
| MLP | 62.29 ± 1.03 | 42.66 ± 0.84 | 38.66 ± 4.02 | 36.74 ± 1.80 | 36.70 ± 0.85 | 89.82 ± 0.63 | 93.57 ± 0.55 | 78.94 ± 1.22 | 87.48 ± 0.46 | 89.96 ± 1.22 | 11 |
| GCN | 38.58 ± 2.35 | 45.16 ± 0.49 | 42.12 ± 3.82 | 38.47 ± 1.82 | 30.11 ± 0.74 | 68.25 ± 2.75 | 78.15 ± 0.95 | 77.53 ± 1.41 | 87.70 ± 0.32 | 94.31 ± 0.33 | 10.8 |
| GAT | 59.55 ± 1.45 | 46.90 ± 0.47 | 40.89 ± 3.50 | 38.22 ± 1.71 | 30.94 ± 0.95 | 57.22 ± 3.04 | 88.36 ± 1.37 | 76.69 ± 0.87 | 87.45 ± 0.53 | 94.59 ± 0.48 | 11.4 |
| GCNII | 82.53 ± 0.37 | 47.53 ± 0.72 | 41.56 ± 4.15 | 40.70 ± 1.80 | **37.51 ± 0.92** | _91.64 ± 0.67_ | _96.48 ± 0.62_ | 84.63 ± 0.66 | 89.96 ± 0.43 | 95.18 ± 0.39 | 4.1 |
| H2GCN | 68.61 ± 1.05 | 37.20 ± 0.67 | 42.29 ± 4.57 | 35.82 ± 2.20 | 33.32 ± 0.90 | 91.25 ± 0.58 | 96.24 ± 0.39 | 78.34 ± 2.01 | 89.32 ± 0.37 | _95.66 ± 0.26_ | 8.2 |
| MixHop | 79.16 ± 0.70 | 47.95 ± 0.65 | 44.97 ± 3.12 | 40.43 ± 1.40 | 36.97 ± 0.90 | 91.10 ± 0.46 | 96.21 ± 0.42 | 84.19 ± 0.61 | 89.42 ± 0.37 | 95.63 ± 0.30 | 4.7 |
| GBK-GNN | 66.05 ± 1.44 | 40.20 ± 1.96 | 42.01 ± 4.89 | 36.52 ± 1.45 | 35.70 ± 1.12 | OOM | OOM | 81.07 ± 0.83 | 88.18 ± 0.45 | 93.48 ± 0.42 | 10.7 |
| GGCN | OOM | OOM | 41.23 ± 4.08 | 36.76 ± 2.19 | 35.68 ± 0.87 | 90.84 ± 0.65 | 95.58 ± 0.44 | _84.76 ± 0.65_ | 89.04 ± 0.40 | 95.18 ± 0.44 | 8.5 |
| GloGNN | 68.63 ± 0.63 | 48.62 ± 0.59 | 40.95 ± 5.95 | 36.85 ± 1.97 | 36.66 ± 0.81 | 90.47 ± 0.77 | 94.51 ± 0.49 | 82.83 ± 0.52 | 89.60 ± 0.34 | 95.09 ± 0.46 | 8.2 |
| HOGGCN | OOM | OOM | 43.35 ± 3.66 | 38.63 ± 1.95 | 36.47 ± 0.83 | 90.94 ± 0.72 | 94.75 ± 0.65 | 83.74 ± 0.69 | OOM | 94.79 ± 0.26 | 7.3 |
| GPR-GNN | 71.19 ± 0.75 | 46.64 ± 0.52 | 41.84 ± 4.68 | 38.04 ± 1.98 | 36.21 ± 0.98 | 91.19 ± 0.47 | 96.37 ± 0.44 | 84.07 ± 0.54 | 89.28 ± 0.37 | 95.48 ± 0.24 | 6.7 |
| ACM-GCN | 71.15 ± 0.73 | 50.64 ± 0.61 | _45.20 ± 4.14_ | _40.90 ± 1.74_ | 35.88 ± 1.40 | 91.43 ± 0.65 | 96.19 ± 0.45 | 84.39 ± 0.43 | 89.99 ± 0.40 | 95.52 ± 0.40 | 4.3 |
| OrderedGNN | _83.10 ± 0.75_ | _51.30 ± 0.61_ | 42.07 ± 4.24 | 37.75 ± 2.53 | _37.22 ± 0.62_ | 91.42 ± 0.79 | 96.27 ± 0.73 | **85.50 ± 0.80** | **90.09 ± 0.37** | **95.73 ± 0.33** | _3.3_ |
| **CMGNN** | **84.35 ± 1.27** | **52.13 ± 0.55** | **45.70 ± 4.92** | **41.89 ± 2.34** | 36.82 ± 0.78 | **92.66 ± 0.46** | **97.00 ± 0.52** | 84.50 ± 0.73 | _89.99 ± 0.32_ | 95.48 ± 0.29 | **2.1** |

**Objective Function.** As mentioned in Sec 4, the CMs in real-world graphs don't always have significant discriminability, which may lead to low effectiveness of supplementary messages. Thus, we introduce an additional discrimination loss $\mathcal{L}_{dis}$ to reduce the similarity of the desired neighborhood message among different classes, which enhances the discriminability among classes in CM. The overall loss consists of a CrossEntropy loss $\mathcal{L}_{ce}$ and the discrimination loss $\mathcal{L}_{dis}$:

$$\mathcal{L} = \mathcal{L}_{ce} + \lambda \mathcal{L}_{dis}, \quad \mathcal{L}_{dis} = \sum_{i \neq j} \text{Sim}(\hat{\mathbf{M}}_i \mathbf{Z}_{ptt}, \hat{\mathbf{M}}_j \mathbf{Z}_{ptt}), \quad (11)$$

where $\mathbf{Z}_{ptt} \in \mathbb{R}^{K \times d_r}$ is the representation of virtual prototypes nodes. More details about the implementation of CMGNN is available in Appendix E.

# 6 Benchmarks and Experiments

In this section, we conduct comprehensive experiments to demonstrate the effectiveness of the proposed CMGNN with a newly organized benchmark for fair comparisons.

## 6.1 New Benchmark

As reported in [30], some widely adopted datasets in existing works have critical drawbacks, which lead to unreliable results. Therefore, with a comprehensive review of existing benchmark evaluation, we construct a new benchmark to fairly perform experimental validation. Specifically, we integrate 13 representative homophilous and heterophilous GNNs, construct a unified codebase, and evaluate their node classification performances on 10 unified organized datasets with various heterophily levels.

**Drawbacks of Existing Datasets.** Existing works mostly follow the settings and datasets used in [15], including 6 heterophilous datasets (Cornell, Texas, Wisconsin, Actor, Chameleon, and Squirrel) and 3 homophilous datasets (Cora, Citeseer, and Pubmed). Platonov et al. [30] pointed out that there are serious data leakages in Chameleon and Squirrel, while Cornell, Texas, and Wisconsin are too small with very imbalanced classes. Further, we revisit other datasets and discover new drawbacks: (i) In the ten splits of Citeseer, there are two inconsistent ones, which have smaller training, validation, and test sets that could cause issues with statistical results; (ii) The data split ratios for Cora are not consistent with the expected ones. These drawbacks may lead to certain issues with the conclusions of previous works. The detailed descriptions of dataset drawbacks are listed in Appendix F.1.

**Newly Organized Datasets.** The datasets used in the benchmark include Roman-Empire, Amazon-Ratings, Chameleon-F, Squirrel-F, Actor, Flickr, BlogCatalog, Wikics, Pubmed, and Photo. Their statistics are summarized in Table 2, with details in Appendix F.2. For consistency with existing methods, we randomly construct 10 splits with predefined proportions (48%/32%/20% for train/valid/test) for each dataset and report the mean performance and standard deviation of 10 splits.

Table 3: Ablation study results (%) between CMGNN and three ablation variants, where SM denotes supplementary messages of the desired neighborhoods and DL denotes the discrimination loss.

| Variants | Roman-Empire | Amazon-Ratings | Chameleon-F | Squirrel-F | Actor | Flickr | BlogCatalog | Wikics | Pubmed | Photo |
|---|---|---|---|---|---|---|---|---|---|---|
| CMGNN | **84.35 ± 1.27** | **52.13 ± 0.55** | **45.70 ± 4.92** | **41.89 ± 2.34** | **36.82 ± 0.78** | **92.66 ± 0.46** | **97.00 ± 0.52** | **84.50 ± 0.73** | **89.99 ± 0.32** | **95.48 ± 0.29** |
| W/O SM | 83.84 ± 1.09 | 51.98 ± 0.61 | 42.35 ± 4.21 | 40.79 ± 1.89 | 36.02 ± 1.21 | 92.32 ± 0.83 | 96.52 ± 0.63 | 83.97 ± 0.83 | 89.70 ± 0.44 | 95.41 ± 0.40 |
| W/O DL | 83.68 ± 1.24 | 52.04 ± 0.37 | 44.97 ± 3.99 | 41.60 ± 2.43 | 36.28 ± 1.12 | **92.66 ± 0.46** | **97.00 ± 0.52** | 83.29 ± 1.83 | **89.99 ± 0.32** | 95.26 ± 0.35 |
| W/O SM and DL | 83.52 ± 1.91 | 51.58 ± 1.04 | 41.12 ± 2.93 | 40.07 ± 2.41 | 35.61 ± 1.48 | 92.32 ± 0.83 | 96.52 ± 0.63 | 81.62 ± 1.67 | 89.70 ± 0.44 | 94.66 ± 0.42 |

**Baseline Methods.** As baseline methods, we choose 13 representative homophilous and heterophilous GNNs, including (i) shallow base model: MLP; (ii) homophilous GNNs: GCN [1], GAT [28], GCNII [27]; (iii) heterophilous GNNs: H2GCN [12], MixHop [16], GBK-GNN [24], GGCN [23], GloGNN [19], HOGGCN [17], GPR-GNN [20], ACM-GCN [18] and OrderedGNN [21]. For each method, we integrate its official/reproduced code into a unified codebase and search for parameters in the space suggested by the original papers. More experimental settings can be found in Appendix F.4 and G.1.

## 6.2 Main Results

Following the constructed benchmark, we evaluate methods and report the performance in Table 2.

**Performance of Baseline Methods.** With the new benchmarks, some interesting observations and conclusions can be found when analyzing the performance of baseline methods. First, comparing the performance of MLP and GCN, we can find "good" heterophily in Amazon-Ratings, Chameleon-F, and Squirrel-F. Meanwhile, when the homophily level is not high enough, "bad" homophily may also exist as shown in BlogCatalog and Wikics. These results once again support the observations about CMs. Therefore, **homophilous GNNs** can also work well in heterophilous graphs as GCNII has an average rank of 4.1, which is better than most HTGNNs. This is attributed to the initial residual connection in GCNII actually playing the role of ego/neighbor separation, which is suitable in heterophilous graphs. As for **heterophilous GNNs**, they are usually designed for both homophilous and heterophilous graphs. Surprisingly, MixHop, as an early method, demonstrated quite good performance. In fact, from the perspective of HTMP, it can be considered a degenerate version of OrderedGNN with no learnable dimensions. As previous SOTA methods, OrderedGNN and ACM-GCN prove their strong capabilities again.

**Performance of CMGNN.** CMGNN achieves the best performance in 6 datasets and an average rank of 2.1, which outperforms baseline methods. This demonstrates the superiority of utilizing and enhancing the CM to handle incomplete and noisy semantic neighborhoods, especially in heterophilous graphs. Regarding the suboptimal performance in Actor, we believe that this is due to the CM in this dataset are not discriminative enough to provide valuable information via the supplementary messages and hard to enhance. In homophilous graphs, due to the identity-like CMs, the overlap between distributions is relatively less, leading to a minor contribution from supplement messages. Yet CMGNN still achieves top-level performances.

## 6.3 Ablation Study

We conduct an ablation study on two key designs of CMGNN , including the supplementary messages of the desired neighborhood (SM) and the discrimination loss (DL). The results are shown in Table 3. *First of all*, both SM and DL have indispensable contributions except for Flickr, BlogCatalog, and Pubmed, in which the discrimination loss has no effect. This may be due to the discriminability of desired neighborhood messages reaching the bottlenecks and can not be further improved by DL *Meanwhile*, the extent of their contributions varies across datasets. SM plays a more important role in most datasets except Roman-Empire, Wikics, and Photo, in which the number of nodes that need supplementary messages is relatively small and DL has great effects. **Further**, we notice that with SM and DL, CMGNN can reach a smaller standard deviation most of the time. This illustrates that CMGNN achieves more stable results by handling nodes with incomplete and noisy semantic neighborhoods. As for the opposite result on Chameleon-F, this may attributed to the small size of this dataset (890 nodes), which can lead to naturally unstable results.

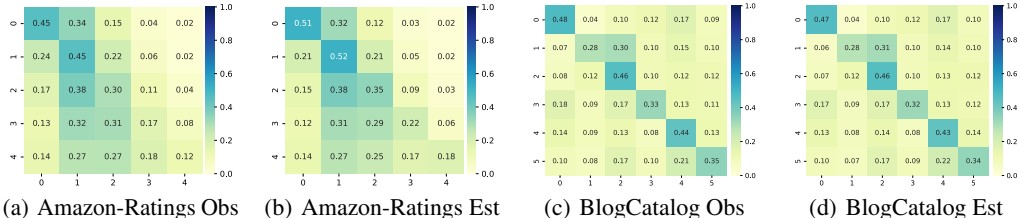

| (a) Amazon-Ratings Obs | (b) Amazon-Ratings Est | (c) BlogCatalog Obs | (d) BlogCatalog Est |

Figure 2: The visualization of observed (Obs) and estimated (Est) compatibility matrixes.

Table 4: Node classification accuracy (%) comparison among nodes with different degrees.

| Dataset | Amazon-Ratings | | | | | Flickr | | | | | BlogCatalog | | | | |
|---|---|---|---|---|---|---|---|---|---|---|---|---|---|---|---|
| Deg. Prop.(%) | 0~20 | 20~40 | 40~60 | 60~80 | 80~100 | 0~20 | 20~40 | 40~60 | 60~80 | 80~100 | 0~20 | 20~40 | 40~60 | 60~80 | 80~100 |
| **CMGNN** | **59.78** | **58.36** | **53.08** | 41.74 | 47.86 | **92.56** | **91.19** | 92.71 | **93.24** | 93.65 | **94.13** | **97.17** | **98.29** | **97.99** | **97.47** |
| ACM-GCN | 57.35 | 56.21 | 51.74 | 41.55 | 46.47 | 90.44 | 91.17 | **92.85** | 93.19 | 89.50 | 92.17 | 96.68 | 97.83 | 97.84 | 96.51 |
| OrderedGNN | 56.32 | 56.16 | 51.20 | **41.85** | **50.26** | 86.48 | 90.07 | 92.40 | 92.79 | 93.40 | 92.19 | 96.09 | 97.48 | 97.36 | 96.27 |
| GCNII | 50.61 | 49.94 | 47.49 | **41.85** | 47.76 | 87.49 | 90.54 | 92.29 | 92.68 | **95.09** | 92.81 | 96.73 | 97.58 | 97.90 | 97.43 |

## 6.4 Visualization of Compatibility Matrix Estimation

We visualize the observed and estimated CMs by CMGNN in Figure 2 with heat maps. Obviously, CMGNN estimates CMs that are very close to those existing in graphs. This shows that even with incomplete node labels, CMGNN can estimate high-quality CMs which provides valuable neighborhood information to nodes. Meanwhile, it can adapt to graphs with various levels of heterophily. More results can be seen in Appendix G.2.1.

## 6.5 Performance on Nodes with Various Levels of Degrees

To verify the effect of CMGNN on nodes with incomplete and noisy semantic neighborhoods, we divide the test set nodes into 5 parts according to their degrees and report the classification accuracy respectively. We compare CMGNN with 3 top-performance methods and show the results in Table 4. In general, nodes with low degrees tend to have incomplete and noisy semantic neighborhoods. Thus, our outstanding performances on the top 20% nodes with the least degree demonstrate the effectiveness of CMGNN for providing desired neighborhood messages. Further, we can find that OrderedGNN and GCNII are good at dealing with nodes with high degrees, while ACM-GCN is relatively good at nodes with low degrees. And CMGNN , to a certain extent, can be adapted to both situations at the same time.

## 7 Conclusion and Limitations

In this paper, we revisit the message passing mechanism in existing heterophilous GNNs and reformulate them into a unified heterophilous message passing (HTMP) mechanism. Based on the HTMP mechanism and empirical analysis, we reveal that the reason for message passing remaining effective is attributed to implicitly enhancing the compatibility matrix among classes. Further, we propose a novel method CMGNN to unlock the potential of the compatibility matrix by handling the incomplete and noisy semantic neighborhoods. The experimental results show the effectiveness of CMGNN and the feasibility of designing a new method following HTMP mechanism. We hope the HTMP mechanism and benchmark can further provide convenience to the community.

This work mainly focuses on the message passing mechanism in existing HTGNNs under the semi-supervised setting. Thus, the other designs in HTGNNs such as objective functions are not analyzed in this paper. The proposed HTMP mechanism is suitable for only a large part of existing HTGNNs which still follow the message passing mechanism.

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

# A More Details of HTMP Mechanism

In this part, we list more details about the HTMP mechanism, including additional analysis about HTMP, method-wise analysis and overall analysis.

## A.1 Additional Analysis of HTMP Mechanism

### A.1.1 Neighborhood Indicators

The neighborhood indicator explicitly marks the neighbors of all nodes within a specific neighborhood. In existing heterophilous GNNs, neighborhood indicators typically take one of the following forms: (i) Raw Neighborhood (Raw); (ii) Neighborhood Redefining (ReDef); and (3) Neighborhood Discrimination (Dis).

**Raw Neighborhood.** Raw neighborhood, including $\mathbf{A}$ and $\tilde{\mathbf{A}}$, provides the basic neighborhood information. The only difference between them lies in whether there is differential treatment of the node's ego messages. For example, APPNP [26] applies additional weighting to the nodes' ego messages compared with GCN [1]. For the sake of simplicity, we consider the identity matrix $\mathbf{I} \in \mathbb{R}^{N \times N}$ as a special neighborhood indicator for acquiring the nodes' ego messages. In heterophilous GNNs, ego/neighbor separation is a common strategy that can mitigate the confusion of ego messages with neighbor messages.

**Neighborhood Redefining.** Neighborhood redefining is the most commonly used technique in heterophilous GNNs, aiming to capture additional information from new neighborhoods. As a representative example, *high-order neighborhood* $\mathbf{A}_h$ can provide long-distance connection information but also result in additional computational costs. *Feature-similarity-based neighborhood* $\mathbf{A}_f$ is often defined by the k-NN relationships within the feature space. Fundamentally, it only utilizes node features and thus needs to be used in conjunction with other neighborhood indicators. Otherwise, the model will be limited by the amount of information in node features. GloGNN [19] introduces *fully-connected neighborhood* $\mathbf{1} \in \mathbb{R}^{N \times N}$, which can capture global neighbor information from all nodes. However, it can also cause significant time and space consumption. Additionally, there are some *custom-defined neighborhood* $\mathbf{A}_c$. For example, Geom-GCN [15] redefines neighborhoods based on the geometric relationships between node pairs. These neighborhood indicators may have limited generality, and the effectiveness is reliant on the specific method.

**Neighborhood Discrimination.** Neighborhood discrimination aims to mark whether neighbors share the same label with central nodes. The neighborhoods are partitioned into positive $\mathbf{A}_p$ and negative ones $\mathbf{A}_n$, which include homophilous and heterophilous neighbors respectively. GGCN [23] divides the raw neighborhood based on the similarity of node representations with a threshold of 0. Explicitly distinguishing neighbors allows for targeted processing, making the model more interpretable. However, its performance is influenced by the accuracy of the discrimination, which may lead to the accumulation of errors.

### A.1.2 Aggregation Guidance

After identifying the neighborhood, the aggregation guidance controls what type of messages to gather from the corresponding neighbors. The existing aggregation guidance mainly includes three kinds of approaches: (1) Degree Averaging (DegAvg), (2) Adaptive Weights (AdaWeight), and (3) Relationship Estimation (RelaEst).

**Degree Averaging.** Degree averaging, formatted as $\mathbf{B}^d = \mathbf{D}^{-\frac{1}{2}} \mathbf{1} \mathbf{D}^{-\frac{1}{2}}$ or $\mathbf{B}^d = \mathbf{D}^{-1} \mathbf{1}$, is the most common aggregation guidance, which plays the role of a low-pass filter to capture the smooth signals and is fixed during model training. Further, combining negative degree averaging with an identity aggregation guidance $\mathbf{I} \in \mathbb{R}^{N \times N}$ can capture the difference between central nodes and neighbors, as used in ACM-GCN [18]. Degree averaging is simple and efficient but depends on the discriminability of corresponding neighborhoods.

**Adaptive Weights.** Another common strategy is allowing the model to learn the appropriate aggregation guidances $\mathbf{B}^{aw}$. GAT [28] proposes an attention mechanism to learn aggregate weights, which guides many subsequent heterophilous methods. To better handle heterophilous graphs, FAGCN [11] introduces negative-available attention weights $\mathbf{B}^{naw}$ to capture the difference between central nodes

and heterophilous neighbors. Adaptive weights can personalize message aggregation for different neighbors, yet it's difficult for models to attain the desired effect.

**Relationship Estimation.** Recently, some methods have tried to estimate the pair-wise relationships $\mathbf{B}^{re}$ between nodes and use them to guide message aggregation. HOG-GCN [17] estimates the pair-wise homophily levels between nodes as aggregation guidances based on both attribute and topology space. GloGNN [19] treats all nodes as neighbors and estimates a coefficient matrix as aggregation guidance based on the idea of linear subspace expression. GGCN [23] estimates appropriate weights for message aggregation with the degrees of nodes and the similarities between node representations. Relationship estimation usually has theoretical guidance, which brings strong interpretability. However, it may also result in significant temporal and spatial complexity when estimating pair-wise relations.

### A.1.3 COMBINE Function

After message aggregation, the COMBINE functions integrate messages from multiple neighborhoods into layer representations. COMBINE functions in heterophilous GNNs are commonly based on two operations: addition and concatenation, each of which has variants. To merge several messages together, addition (Add) is a naive idea. Further, to control the weight of messages from different neighborhoods, weighted addition (WeightedAdd) is applied. However, it is a global setting and cannot adapt to the differences between nodes. Thus, adaptive weighted addition (AdaAdd) is proposed, which can learn personalized message combination weights for each node, but it will result in additional time consumption. Although the addition is simple and efficient, some methods [12, 16] believe that it may blur messages from different neighborhoods, which can be harmful in heterophilous GNNs, so they employ a concatenation operation (Cat) to separate the messages. Nevertheless, such an approach not only increases the space cost but may also retain additional redundant messages. To address these issues, OrderedGNN [21] proposes an adaptive concatenation mechanism (AdaCat) that can combine multiple messages with learnable dimensions. This is an innovative and worthy further exploration practice, but the difficulty of model learning should also be considered.

### A.1.4 FUSE Function

Further, the FUSE functions integrate messages from multiple layers into the final representation. For the FUSE function, utilizing the representation of the last layer as the final representation is widely accepted: $\mathbf{Z} = \mathbf{Z}^L$. JKNet [31] proposes that the combination of representations from intermediate layers can capture both local and global information. H2GCN [12] applies it in heterophilous graphs, preserving messages from different localities with concatenation. Similarly, GPRGNN [20] combines the representations of multiple layers into the final representation through adaptive weighted addition.

### A.1.5 AGGREGATE function

The most commonly used AGGREGATE function is $\mathbf{AGGREGATE}(\mathbf{A}_r, \mathbf{B}_r, \mathbf{Z}_r^{l-1}) = (\mathbf{A}_r \odot \mathbf{B}_r)\mathbf{Z}_r^{l-1}\mathbf{W}_r^l$. We take this as the fixed form of the AGGREGATE function following. Actually, the input representations $\mathbf{Z}_r^{-1}$ and weight matrixes $\mathbf{W}_r^l$ also can be specially designed. Taking the initial node representations $\mathbf{Z}^0$ as input is a relatively common approach as in APPNP [26], GCNII [27], FAGCN [11] and GloGNN [19]. Further, GCNII [27] adds an identity matrix $\mathbf{I}_w$ to the weight matrixes to keep more original messages. However, the methods that specially design these components are few and with a similar form. Thus, we don't discuss them too much, but leave it for future extensions.

## A.2 Revisiting Representative GNNs with HTMP Mechanism

In this part, we utilize HTMP mechanism to revisit the representative GNNs. We start from homophilous GNNs as simple examples and further extend to heterophilous GNNs.

### A.2.1 GCN

Graph Convolutional Networks (GCN) [1] utilizes a low-pass filter to gather messages from neighbors as follows:
$$\mathbf{Z}^l = \hat{\hat{\mathbf{A}}}\mathbf{Z}^{l-1}\mathbf{W}^l. \tag{12}$$

It can be revisited by HTMP with the following components:

$$\mathbf{A}_0 = \tilde{\mathbf{A}}, \quad \mathbf{B}_0 = \mathbf{B}^d = \tilde{\mathbf{D}}^{-\frac{1}{2}} \mathbf{1} \tilde{\mathbf{D}}^{-\frac{1}{2}},$$
$$\mathbf{Z}^l = \mathbf{Z}_0^l = (\mathbf{A}_0 \odot \mathbf{B}_0)\mathbf{Z}^{l-1}\mathbf{W}^l = \hat{\tilde{\mathbf{A}}}\mathbf{Z}^{l-1}\mathbf{W}^l. \tag{13}$$

Specifically, GCN has a raw neighborhood indicator $\tilde{\mathbf{A}}$ and a degree averaging aggregation guidance $\mathbf{B}^d$. Since there is only one neighborhood, the COMBINE function is meaningless in GCN. GCN utilizes a naive way to fuse messages about the original neighborhood and central nodes. However, it may confuse the representations in heterophilous graphs.

### A.2.2 APPNP

PPNP [26] is also a general method whose message passing is based on Personalized PageRank (PPR). To avoid massive consumption, APPNP is introduced as the approximate version of PPNP with an iterative message-passing mechanism:

$$\mathbf{Z}^l = \mu\mathbf{Z}^0 + (1 - \mu)\hat{\mathbf{A}}\mathbf{Z}^{l-1}. \tag{14}$$

It can be revisited by _with the following components:

$$\mathcal{A} = [\mathbf{A}_0, \ \mathbf{A}_1], \quad \mathcal{B} = [\mathbf{B}_0, \ \mathbf{B}_1],$$
$$\mathbf{A}_0 = \mathbf{I}, \quad \mathbf{B}_0 = \mathbf{I}, \quad \mathbf{W}_0^l = \mathbf{I},$$
$$\widetilde{\mathbf{Z}}_0^l = (\mathbf{A}_0 \odot \mathbf{B}_0)\mathbf{Z}^0\mathbf{W}_0^l = \mathbf{Z}^0, \tag{15}$$
$$\mathbf{A}_1 = \mathbf{A}, \quad \mathbf{B}_1 = \mathbf{D}^{-\frac{1}{2}}\mathbf{1}\mathbf{D}^{-\frac{1}{2}}, \quad \mathbf{W}_1^l = \mathbf{I},$$
$$\widetilde{\mathbf{Z}}_1^l = (\mathbf{A}_1 \odot \mathbf{B}_1)\mathbf{Z}^{l-1}\mathbf{W}_1^l = \hat{\mathbf{A}}\mathbf{Z}^{l-1}.$$

Specifically, APPNP aggregates messages from node ego and neighborhoods separately and combines them with a weighted addition. Compared with GCN, APPNP assigns adjustable weights to nodes, for controlling the proportion of ego and neighbor messages during message-passing, which becomes a worthy design in heterophilous graphs.

### A.2.3 GAT

Going a step further, Graph Attention Networks (GAT) [28] allows learnable weights for each neighbor:

$$\mathbf{Z}_i^l = \sum_{j \in \tilde{\mathcal{N}}(i)} \alpha_{ij}\mathbf{Z}_j^{l-1}\mathbf{W}^l, \tag{16}$$

where $\alpha_{ij}$ is the weight for aggregating neighbor node $j$ to center node $i$, whose construction process is as follows:

$$\alpha_{ij} = \frac{\exp(e_{ij})}{\sum_{k \in \tilde{\mathcal{N}}(i)} \exp(e_{ik})}, \tag{17}$$
$$e_{ij} = \text{LeakyReLU}\left(\left[\mathbf{Z}_i^{l-1} | \mathbf{Z}_j^{l-1}\right]\mathbf{a}\right).$$

Let $\mathbf{P}^{GAT}$ be the matrix of aggregation weights in GAT:

$$\mathbf{P}_{ij}^{GAT} = \begin{cases} \alpha_{ij}, & \tilde{\mathbf{A}}_{ij} = 1, \\ 0, & \tilde{\mathbf{A}}_{ij} = 0. \end{cases} \tag{18}$$

HTMP can revisit GAT with the following components:

$$\mathbf{A}_0 = \tilde{\mathbf{A}}, \quad \mathbf{B}_0 = \mathbf{B}^{aw} = \mathbf{P}^{GAT},$$
$$\mathbf{Z}^l = \mathbf{Z}_0^l = (\mathbf{A}_0 \odot \mathbf{B}_0)\mathbf{Z}^{l-1}\mathbf{W}^l = \mathbf{P}^{GAT}\mathbf{Z}^{l-1}\mathbf{W}^l, \tag{19}$$

which is the matrix version of Eq 16. Specifically, GAT aggregate messages from raw neighborhood $\tilde{\mathbf{A}}$ with adaptive weights $\mathbf{B}^{aw}$. Aggregation guidance with adaptive weights is a nice idea, but simple constraints are not enough for the model to learn ideal results.

## A.2.4 GCNII

GCNII [27] is a novel homophilous GNN with two key designs: initial residual connection and identity mapping, which can be formatted as follows:

$$\mathbf{Z}^l = \left(\alpha \mathbf{Z}^0 + (1-\alpha)\tilde{\mathbf{D}}^{-\frac{1}{2}}\tilde{\mathbf{A}}\tilde{\mathbf{D}}^{-\frac{1}{2}}\mathbf{Z}^{l-1}\right)\left(\beta \mathbf{W}^l + (1-\beta)\mathbf{I}_w\right), \tag{20}$$

where $\alpha$ and $\beta$ are two predefined parameters and $\mathbf{I}_w \in \mathbb{R}^{d_r \times d_r}$ is an identity matrix.

From the perspective of HTMP, it can be viewed as follows:

$$\begin{aligned}
\mathcal{A} &= [\mathbf{I}, \tilde{\mathbf{A}}], \quad \mathcal{B} = [\mathbf{I}, \tilde{\mathbf{B}}^d], \quad \mathbf{W}_0^l = \mathbf{W}_1^l = \left(\beta\mathbf{W}^l + (1-\beta)\mathbf{I}_w\right), \\
\widetilde{\mathbf{Z}}_0^l &= (\mathbf{I} \odot \mathbf{I})\mathbf{Z}^0\left(\beta\mathbf{W}^l + (1-\beta)\mathbf{I}_w\right) = \mathbf{Z}^0\left(\beta\mathbf{W}^l + (1-\beta)\mathbf{I}_w\right), \\
\widetilde{\mathbf{Z}}_1^l &= (\tilde{\mathbf{A}} \odot \tilde{\mathbf{B}}^d)\mathbf{Z}^{l-1}\left(\beta\mathbf{W}^l + (1-\beta)\mathbf{I}_w\right) = \hat{\tilde{\mathbf{A}}}\mathbf{Z}^{l-1}\left(\beta\mathbf{W}^l + (1-\beta)\mathbf{I}_w\right),
\end{aligned} \tag{21}$$

where the COMBINE function is weighted addition. Specifically, the first design of GCNII is a form of ego/neighbor separation, and the second design is a novel transformation weights matrix. This can also be specially designed, but only GCNII does this, so we won't analyze it too much and leave it as a future extension.

## A.2.5 Geom-GCN

Geom-GCN [15] is one of the most influential heterophilous GNNs, which employs the geometric relationships of nodes within two kinds of neighborhoods to aggregate the messages through bi-level aggregation:

$$\begin{aligned}
\mathbf{Z}^l &= \left(\underset{i \in \{g,s\}}{\|} \underset{r \in R}{\|} \mathbf{Z}_{i,r}^l\right)\mathbf{W}^l, \\
\mathbf{Z}_{i,r}^l &= \mathbf{D}_{i,r}^{-\frac{1}{2}}\mathbf{A}_{i,r}\mathbf{D}_{i,r}^{-\frac{1}{2}}\mathbf{Z}^{l-1},
\end{aligned} \tag{22}$$

where $\|$ denotes the concatenate operator, $\{g, s\}$ is the set of neighborhoods including the original graph and the latent space. $R$ is the set of geometric relationships. $\mathbf{A}_{i,r}$ is the corresponding adjacency matrix in neighborhood $i$ and relationship $r$.

It can be revisited by HTMP with the following components:

$$\begin{aligned}
\mathcal{A} &= [\mathbf{A}_{i,r}|i \in \{g,s\}, r \in R], \quad \mathcal{B} = [\mathbf{B}_{i,r}^d\|i \in \{g,s\}, r \in R], \\
\widetilde{\mathbf{Z}}_{i,r}^l &= (\mathbf{A}_{i,r} \odot \mathbf{B}_{i,r}^d)\mathbf{Z}_{l-1}\mathbf{W}_{i,r}^l = \mathbf{D}_{i,r}^{-\frac{1}{2}}\mathbf{A}_{i,r}\mathbf{D}_{i,r}^{-\frac{1}{2}}\mathbf{Z}^{l-1}\mathbf{W}_{i,r}^l,
\end{aligned} \tag{23}$$

where the COMBINE function is concatenation and the weight matrix $\mathbf{W}^l$ in Eq 22 can be viewed as the combination of multiple $\mathbf{W}_{i,r}^l$. Specifically, Geom-GCN redefines multiple neighborhoods based on the customized geometric relations in both raw and latent space. The messages are aggregated from each neighborhood and combined by a concatenation. This approach may be applicable to some datasets, yet it has weak universality.

## A.2.6 H2GCN

H2GCN [12] is also an influential method with three key designs: ego- and neighbor-message separation, higher-order neighborhoods, and the combination of intermediate representations. Its single-layer representations are constructed as follows:

$$\mathbf{Z}^l = \left[\hat{\mathbf{A}}\mathbf{Z}^{l-1} \| \hat{\mathbf{A}}_{h2}\mathbf{Z}^{l-1}\right], \tag{24}$$

where $\hat{\mathbf{A}}_{h2}$ denotes the 2-order adjacency matrix with normalization.

It can be revisited by HTMP with the following components:

$$\begin{aligned}
\mathcal{A} &= [\mathbf{A}, \mathbf{A}_{h2}], \quad \mathcal{B} = [\mathbf{B}^d, \mathbf{B}_{h2}^d], \quad \mathbf{W}_0^l = \mathbf{W}_1^l = \mathbf{I}, \\
\widetilde{\mathbf{Z}}_0^l &= (\mathbf{A} \odot \mathbf{B}^d)\mathbf{Z}^{l-1}\mathbf{I} = \hat{\mathbf{A}}\mathbf{Z}^{l-1}, \\
\widetilde{\mathbf{Z}}_1^l &= (\mathbf{A}_{h2} \odot \mathbf{B}_{h2}^d)\mathbf{Z}^{l-1}\mathbf{I} = \hat{\mathbf{A}}_{h2}\mathbf{Z}^{l-1},
\end{aligned} \tag{25}$$

where the COMBINE function is concatenation. Meanwhile, H2GCN also uses the concatenation as the FUSE function. Specifically, H2GCN aggregates messages from the raw and 2-order neighborhoods in a layer of message passing and keeps them apart in the representations. The design of ego/neighbor separation is first introduced by H2GCN and gradually becomes a necessity for subsequent methods.

### A.2.7 SimP-GCN

SimP-GCN [14] constructs an additional graph based on the feature similarity. It has two key concepts: (1) the information from the original graph and feature kNN graph should be balanced, and (2) each node can adjust the contribution of its node features. Specifically, the message passing in SimP-GCN is as follows:

$$\mathbf{Z}^l = \left( \text{diag}(\mathbf{s}^l)\hat{\tilde{\mathbf{A}}} + \text{diag}(1 - \mathbf{s}^l)\hat{\mathbf{A}}_f + \gamma \mathbf{D}_K^l \right) \mathbf{Z}^{l-1}\mathbf{W}^l, \tag{26}$$

where $\mathbf{s}^l \in \mathbb{R}^n$ is a learnable score vector that balances the effect of the original and feature graphs, $\mathbf{D}_K^l = \text{diag}(K_1^l, K_2^l, ..., K_n^l)$ is a learnable diagonal matrix.

It can be revisited by HTMP with the following components:

$$\begin{aligned}
\mathcal{A} &= [\mathbf{I}, \tilde{\mathbf{A}}, \mathbf{A}_f], \quad \mathcal{B} = [\mathbf{I}, \tilde{\mathbf{B}}^d, \mathbf{B}_f^d], \\
\widetilde{\mathbf{Z}}_0^l &= (\mathbf{I} \odot \mathbf{I})\mathbf{Z}^{l-1}\mathbf{W}^l = \mathbf{Z}^{l-1}\mathbf{W}^l, \\
\widetilde{\mathbf{Z}}_1^l &= (\tilde{\mathbf{A}} \odot \tilde{\mathbf{B}}^d)\mathbf{Z}^{l-1}\mathbf{W}^l = \hat{\tilde{\mathbf{A}}}\mathbf{Z}^{l-1}\mathbf{W}^l, \\
\widetilde{\mathbf{Z}}_2^l &= (\mathbf{A}_f \odot \mathbf{B}_f^d)\mathbf{Z}^{l-1}\mathbf{W}^l = \hat{\mathbf{A}}_f\mathbf{Z}^{l-1}\mathbf{W}^l,
\end{aligned} \tag{27}$$

where the COMBINE function is adaptive weighted addition. Specifically, SimP-GCN aggregates messages from ego, raw and feature-similarity-based neighborhoods, and combines them with node-specific learnable weights. The feature-similarity-based neighborhoods can provide more homophilous messages to enhance the discriminability of the compatibility matrix. However, it's still limited by the amount of information on node features.

### A.2.8 FAGCN

FAGCN [11] proposes considering both low-frequency and high-frequency information simultaneously, and transferring them into the negative-allowable weights during message passing:

$$\mathbf{Z}_i^l = \mu\mathbf{Z}_i^0 + \sum_{j \in \mathcal{N}_i} \frac{\alpha_{ij}^G}{\sqrt{d_i d_j}}\mathbf{Z}_j^{l-1}, \tag{28}$$

where $\alpha_{ij}^G$ can be negative as follows:

$$\alpha_{ij}^G = \tanh(\mathbf{g}^T[\mathbf{X}_i \| \mathbf{X}_j]), \tag{29}$$

which can form a weight matrix:

$$\mathbf{P}_{ij}^{FAG} = \begin{cases} \alpha_{ij}^G, & \mathbf{A}_{ij} = 1, \\ 0, & \mathbf{A}_{ij} = 0. \end{cases} \tag{30}$$

It can be revisited by HTMP with the following components:

$$\begin{aligned}
\mathcal{A} &= [\mathbf{I}, \mathbf{A}], \quad \mathcal{B} = [\mathbf{I}, \mathbf{D}^{-\frac{1}{2}}\mathbf{P}^{FAG}\mathbf{D}^{-\frac{1}{2}}], \quad \mathbf{W}_0^l = \mathbf{W}_1^l = \mathbf{I}, \\
\widetilde{\mathbf{Z}}_0^l &= (\mathbf{I} \odot \mathbf{I})\mathbf{Z}^0\mathbf{I} = \mathbf{Z}^0, \\
\widetilde{\mathbf{Z}}_1^l &= (\mathbf{A} \odot \mathbf{D}^{-\frac{1}{2}}\mathbf{P}^{FAG}\mathbf{D}^{-\frac{1}{2}})\mathbf{Z}^{l-1}\mathbf{I} = \mathbf{D}^{-\frac{1}{2}}\mathbf{P}^{FAG}\mathbf{D}^{-\frac{1}{2}}\mathbf{Z}^{l-1},
\end{aligned} \tag{31}$$

where the COMBINE function is weighted addition, same as the matrix form of Eq 28. Specifically, FAGCN aggregates messages from node ego and raw neighborhood with negative-allowable weights. It has a similar form to GAT but allows for ego/neighbor separation and negative weights, which means the model can capture the difference between center nodes and neighbors.

 **A.2.9 GGCN**

644 GGCN [23] explicitly distinguishes between homophilous and heterophilous neighbors based on
645 node similarities, and assigns corresponding positive and negative weights:

$$\mathbf{Z}^l = \alpha^l \left( \beta_0^l \hat{\mathbf{Z}}^l + \beta_1^l (\mathbf{S}_{pos}^l \odot \tilde{\mathbf{A}}_{\mathcal{T}}^l) \hat{\mathbf{Z}}^l + \beta_2^l (\mathbf{S}_{neg}^l \odot \tilde{\mathbf{A}}_{\mathcal{T}}^l) \hat{\mathbf{Z}}^l \right), \tag{32}$$

646 where $\hat{\mathbf{Z}}^l = \mathbf{Z}^{l-1}\mathbf{W}^l + b^l$, $\tilde{\mathbf{A}}_{\mathcal{T}}^l = \tilde{\mathbf{A}} \odot \mathcal{T}^l$ is an adjacency matrix weighted by the structure property,
647 $\beta_0^l, \beta_1^l$ and $\beta_2^l$ are learnable scalars. The neighbors are distinguished by the cosine similarity of node
648 representations with a threshold of 0:

$$\mathbf{S}_{ij}^l = \begin{cases} \text{Cosine}(\mathbf{Z}_i, \mathbf{Z}_j), & i \neq j \ \& \ \mathbf{A}_{ij} = 1, \\ 0, & \text{otherwise.} \end{cases},$$

$$\mathbf{S}_{pos,\ ij}^l = \begin{cases} \mathbf{S}_{ij}^l, & \mathbf{S}_{ij}^l > 0, \\ 0, & \text{otherwise.} \end{cases}, \tag{33}$$

$$\mathbf{S}_{neg,\ ij}^l = \begin{cases} \mathbf{S}_{ij}^l, & \mathbf{S}_{ij}^l < 0, \\ 0, & \text{otherwise.} \end{cases}.$$

649 It can be revisited by HTMP with the following components:

$$\begin{aligned}
\mathcal{A} &= [\mathbf{I}, \mathbf{A}_p, \mathbf{A}_n], \quad \mathcal{B} = [\mathbf{I}, \mathbf{S}_{pos}^l \odot \mathcal{T}^l, \mathbf{S}_{neg}^l \odot (T)^l], \\
\widetilde{\mathbf{Z}}_0^l &= (\mathbf{I} \odot \mathbf{I})\mathbf{Z}^{l-1}\mathbf{W}^l = \mathbf{Z}^{l-1}\mathbf{W}^l, \\
\widetilde{\mathbf{Z}}_1^l &= (\mathbf{A}_p \odot \mathbf{S}_{pos}^l \odot \mathcal{T}^l)\mathbf{Z}^{l-1}\mathbf{W}^l = (\mathbf{S}_{pos}^l \odot \mathcal{T}^l)\mathbf{Z}^{l-1}\mathbf{W}^l, \\
\widetilde{\mathbf{Z}}_2^l &= (\mathbf{A}_n \odot \mathbf{S}_{neg}^l \odot \mathcal{T}^l)\mathbf{Z}^{l-1}\mathbf{W}^l = (\mathbf{S}_{neg}^l \odot \mathcal{T}^l)\mathbf{Z}^{l-1}\mathbf{W}^l,
\end{aligned} \tag{34}$$

650 where $\mathbf{A}_p$ and $\mathbf{A}_n$ are discriminated by the representation similarities:

$$\begin{aligned}
\mathbf{A}_{p,ij} &= \begin{cases} 1, & \mathbf{S}_{pos,ij}^l > 0 \& \mathbf{A}_{ij} = 1, \\ 0, & \text{otherwise.} \end{cases}, \\
\mathbf{A}_{n,ij} &= \begin{cases} 1, & \mathbf{S}_{neg,ij}^l < 0 \& \mathbf{A}_{ij} = 1, \\ 0, & \text{otherwise.} \end{cases}.
\end{aligned} \tag{35}$$

651 The COMBINE function is an adaptive weighted addition. Specifically, GGCN divides the raw
652 neighborhood into positive and negative ones based on the similarities among node presentations.
653 On this basis, it aggregates messages from node ego, positive and negative neighborhoods, and
654 combines them with node-specific learnable weights. This approach allows for targeted processing
655 for homophilous and heterophilous neighbors, yet can suffer from the accuracy of discrimination,
656 which may lead to the accumulation of errors.

**A.2.10 ACM-GCN**

658 ACM-GCN [18] introduces 3 channels (identity, low pass and high pass) to capture different informa-
659 tion and mixes them with node-wise adaptive weights:

$$\mathbf{Z}^l = \text{diag}(\alpha_I^l)\mathbf{Z}^{l-1}\mathbf{W}_I^l + \text{diag}(\alpha_L^l)\hat{\mathbf{A}}\mathbf{Z}^{l-1}\mathbf{W}_L^l + \text{diag}(\alpha_H^l)(\mathbf{I} - \hat{\mathbf{A}})\mathbf{Z}^{l-1}\mathbf{W}_H^l, \tag{36}$$

660 where $\text{diag}(\alpha_I^l), \text{diag}(\alpha_L^l), \text{diag}(\alpha_H^l) \in \mathbb{R}^{N \times 1}$ are learnable weight vectors.

661 It can be revisited by HTMP with the following components:

$$\begin{aligned}
\mathcal{A} &= [\mathbf{I}, \mathbf{A}, \mathbf{A}], \quad \mathcal{B} = [\mathbf{I}, \mathbf{B}^d, \mathbf{I} - \mathbf{B}^d], \\
\widetilde{\mathbf{Z}}_0^l &= (\mathbf{I} \odot \mathbf{I})\mathbf{Z}^{l-1}\mathbf{W}_I^l = \mathbf{Z}^{l-1}\mathbf{W}_I^l, \\
\widetilde{\mathbf{Z}}_1^l &= (\mathbf{A} \odot \mathbf{B}^d)\mathbf{Z}^{l-1}\mathbf{W}_L^l = \hat{\mathbf{A}}\mathbf{Z}^{l-1}\mathbf{W}_L^l, \\
\widetilde{\mathbf{Z}}_2^l &= (\mathbf{A} \odot (\mathbf{I} - \mathbf{B}^d))\mathbf{Z}^{l-1}\mathbf{W}_H^l = (\mathbf{I} - \hat{\mathbf{A}})\mathbf{Z}^{l-1}\mathbf{W}_H^l,
\end{aligned} \tag{37}$$

662 where the COMBINE function is adaptive weighted addition. Specifically, ACM-GCN aggregates
663 node ego, low-frequency, and high-frequency messages from ego and raw neighborhoods, and
664 combines them with node-wise adaptive weights. With simple but effective designs, ACM-GCN
665 achieves outstanding performance, which shows that complicated designs are not necessary.

### A.2.11 OrderedGNN

OrderedGNN [21] is a SOTA method that introduces a node-wise adaptive dimension concatenation function to combine messages from neighbors of different hops:

$$\mathbf{Z}^l = \mathbf{P}_d^l \odot \mathbf{Z}^{l-1} + (1 - \mathbf{P}_d^l) \odot (\hat{\mathbf{A}} \mathbf{Z}^{l-1}), \tag{38}$$

where $\mathbf{P}_d \in \mathbb{R}^{N \times d_r}$ is designed to be matrix with each line $\mathbf{P}_{d,i}^l$ being a dimension indicate vector, which starts with continuous 1s while the others be 0s. In practice, to keep the differentiability, it's "soften" as follows:

$$\begin{aligned}
\hat{\mathbf{P}}_d^l &= \text{cumsum}_{\leftarrow}\left(\text{softmax}\left(f_\xi^l\left(\mathbf{Z}^{l-1}, \hat{\mathbf{A}}\mathbf{Z}^{l-1}\right)\right)\right), \\
\mathbf{P}_d^l &= \text{SOFTOR}(\mathbf{P}_d^{l-1}, \hat{\mathbf{P}}_d^l),
\end{aligned} \tag{39}$$

where $f_\xi^l$ is a learnable layer that fuses two messages.

It can be revisited by HTMP with the following components:

$$\begin{aligned}
\mathcal{A} &= [\mathbf{I}, \mathbf{A}], \quad \mathcal{B} = [\mathbf{I}, \mathcal{B}^d], \quad \mathbf{W}_0^l = \mathbf{W}_1^l = \mathbf{I}, \\
\widetilde{\mathbf{Z}}_0^l &= (\mathbf{I} \odot \mathbf{I})\mathbf{Z}^{l-1} = \mathbf{Z}^{l-1}, \\
\widetilde{\mathbf{Z}}_1^l &= (\mathbf{A} \odot \mathbf{B}^d)\mathbf{Z}^{l-1} = \hat{\mathbf{A}}\mathbf{Z}^{l-1},
\end{aligned} \tag{40}$$

where the COMBINE function is concatenation with node-wise adaptive dimensions. Specifically, in each layer, OrderedGNN aggregates messages from node ego and raw neighborhood and concatenates them with learnable dimensions. Combined with the multi-layer architecture, this approach can aggregate messages from neighbors of different hops and combine them not only with adaptive contributions but also as separately as possible.

### A.3 Analysis and Advice for Designing Models

The HTMP mechanism splits the message-passing mechanism of HTGNNs into multiple modules, establishing connections among methods. For instance, most message passing in HTGNNs have personalized processing for nodes. Some methods [24, 11, 13, 22] utilize the learnable aggregation guidance and some others [14, 18, 21, 23] count on learnable COMBINE functions. Though neighborhood redefining is commonly used in HTGNNs, there are also many methods [24, 11, 18, 20, 21] using only raw neighborhoods to handle heterophily and achieve good performance. Degree averaging, which plays the role of a low-pass filter to capture the smooth signals, can still work well in many HTGNNs [12, 14–16, 20]. High-order neighbor information may be helpful in heterophilous graphs. Existing HTGNNs utilize it in two ways: directly defining high-order [12, 13, 16, 17] or even full-connected [19] neighborhood indicators and by the multi-layer architecture of message passing [20, 21].

With the aid of HTMP, we can revisit existing methods from a unified and comprehensible perspective. An obvious observation is that *the coordination among designs is important while good combinations with easy designs can also achieve wonderful results.* For instance, in ACM-GCN [18], the separation and adaptive addition of ego, low-frequency, and high-frequency messages can accommodate the personalized conditions of each node. OrderedGNN's design [21], which includes an adaptive connection mechanism, ego/neighbor separation, and multi-layer architecture, allows discrete and adaptive combinations of messages from multi-hop neighborhoods. This advises us to *take into account all components simultaneously* when designing models. As an illustration, please be cautious about using multiple learnable components. Also, here are some additional model design tips and considerations. Please *separate the messages from node ego and neighbors*. When combining them afterward, whether by weighted addition or concatenation, this approach is at least harmless if not beneficial, especially when dealing with heterophilous graphs. Last but not least, try to design a model capable of *personalized handling different nodes*. Available components include but are not limited to, custom-defined neighborhood indicators, aggregation guidance with adaptive weights or estimated relationships, and learnable COMBINE functions. This is to accommodate the diversity and sparsity of neighborhoods that nodes in real-world graphs may have.

## B    Related Works

**Homophilous Graph Neural Networks**. Graph Neural Networks (GNNs) have showcased impressive capabilities in handling graph-structured data. Traditional GNNs are predominantly founded on the assumption of homophily, broadly categorized into two classes: spectral-based GNNs and spatial-based GNNs. **Firstly**, spectral-based GNNs acquire node representations through graph convolution operations employing diverse graph filters [1, 32, 33]. **Secondly**, spatial-based methods gather information from neighbors and update the representation of central nodes through the message-passing mechanism [26, 28, 34]. Moreover, for a more **comprehensive understanding of existing homophilous GNNs**, several unified frameworks [35, 36] have been proposed. Ma et al. [35] propose that the aggregation process in some representative homophilous GNNs can be regarded as solving a graph denoising problem with a smoothness assumption. Zhu et al. [36] establishes a connection between various message-passing mechanisms and a unified optimization problem. However, these methods have limitations, as the aggregated representations may lose discriminability when heterophilous neighbors dominate [11, 12].

**Heterophilous Graph Neural Networks**. Recently, some heterophilous GNNs have emerged to tackle the heterophily problem [11–23]. **Firstly**, a commonly adopted strategy involves *expanding the neighborhood with higher homophily or richer messages*, such as high order neighborhooods [12, 13], feature-similarity-based neighborhoods [13, 14], and custom-defined neighborhoods [15, 22]. **Secondly**, some approaches [11, 17–19, 23] aim to *leverage information from heterophilous neighbors*, considering that not all heterophily is detrimental et al.[6]. **Thirdly**, some methods [12, 16, 20, 21] adapt to heterophily by extending the combine function in message passing, creating variations for addition and concatenation. On this basis, several works have **reviewed existing heterophilous methods**. Zheng et al. [8] and Zhu et al. [9] identifies effective designs in heterophilous GNNs and analyzes the relationship between heterophily and graph-related issues. Gong et al. [10] provide a higher-level perspective on learning heterophilous graphs, summarizing and classifying existing methods based on learning strategies, architectures, and applications. However, *these reviews merely classify and list methods hierarchically, lacking unified understandings and not exploring the reason behind the effectiveness of message passing in heterophilous graphs.*

## C    The Detail of Experiments on Synthetic Datasets

To explore the performance impact of homophily level, node degrees and compatibility matrix (CMs) on simple GNNs, we conduct some experiments on synthetic datasets.

### C.1    Synthetic Datasets

We construct synthetic graphs considering the factors of homophily, CMs and degrees. For homophily, we set 3 levels including Lowh (0.2), Midh (0.5), and Highh (0.8). For CMs, we set two levels of discriminability, including Easy and Hard. For degrees, we set two levels including Lowdeg (4) and Highdeg (18). Note that with a certain homophily level, we can only control the non-diagonal elements of CMs. Thus, there are a total of 12 synthetic graphs following the above settings. These synthetic graphs are based on the Cora dataset, which provides node features and labels, which means, only the edges are constructed. We visualize the CMs of these graphs in Figure 3. Since there is no significant difference in CMs between low-degree and high-degree, we only plot the high-degree ones. Further, the edges are randomly constructed under the guidance of these CMs and degrees to form the synthetic graphs.

### C.2    Experiments on Synthetic Datasets

We use GCN to analyze the performance impact of the above factors. The semi-supervised node classification performance of GCN is shown in Table 5 while the baseline performance of MLP (72.54 ± 2.18) is the same among these datasets since their difference is only on edges. From these results, we have some observations: (1) high homophily is not necessary, GCN can also work well on low homophily but discriminative CM; (2) low degrees have a negative impact on performance, especially when the CMs are relatively weak discriminative, this also indicates that nodes with lower degrees are more likely to have confused neighborhoods; and (3) when dealing with nodes with confused neighborhoods, GCN may contaminate central nodes with their neighborhoods' messages, which

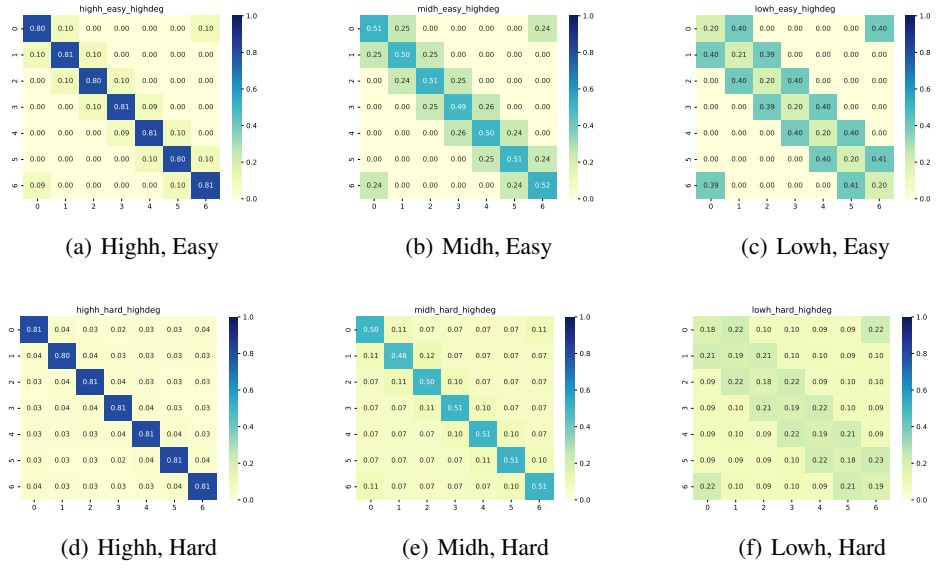

(a) Highh, Easy      (b) Midh, Easy      (c) Lowh, Easy

(d) Highh, Hard      (e) Midh, Hard      (f) Lowh, Hard

Figure 3: The visualization of compatibility matrix on synthetic graphs.

Table 5: Node classification accuracy of GCN on Synthetic Datasets.

| Factors | Highh, Esay | Highh, Hard | Midh, Easy | Midh, Hard | Lowh, easy | Lowh, Hard |
|---|---|---|---|---|---|---|
| **Highd** | $99.15 \pm 0.35$ | $99.48 \pm 0.24$ | $86.42 \pm 4.13$ | $90.52 \pm 1.05$ | $89.34 \pm 2.19$ | $39.22 \pm 2.34$ |
| **Lowd** | $89.98 \pm 1.59$ | $91.25 \pm 0.85$ | $70.85 \pm 1.59$ | $70.20 \pm 1.41$ | $56.46 \pm 2.63$ | $40.91 \pm 1.75$ |

leads to performance worse than MLP. This once again remind us the importance of ego/neighbor separation.

## D    Empirical Evidence for the Conjecture about CM

In this part, we show the empirical evidence for the conjecture about CM as mentioned in Sec 4. Specifically, we plot the observed and desired CM of ACM-GCN and GPRGNN in Figure 4. The results show that ACM-GCN and GPRGNN have enhanced the discriminability of CM, which can be empirical evidence for the conjecture.

The desired CMs are obtained as follows: For ACM-GCN, we leverage the learned weights in the COMBINE function to rebuild a weighted adjacency matrix $\mathbf{A}^{acm}$ based on the low-pass filter $\hat{\mathbf{A}}$ and high-pass filter $\mathbf{I} - \hat{\mathbf{A}}$, then regard $\mathbf{A}^{acm}$ as the neighborhood and calculate the desired CM.

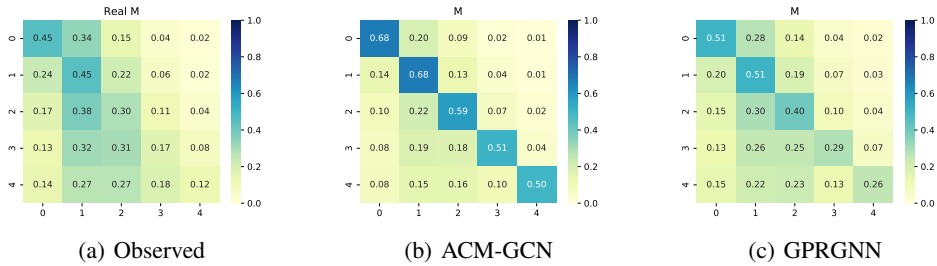

(a) Observed      (b) ACM-GCN      (c) GPRGNN

Figure 4: The visualization of compatibility matrix on Amazon-Ratings.

For GPRGNN, we utilize the leaned weights in the FUSE function to rebuild a weighted adjacency matrix $\mathbf{A}^{gpr}$ based on the multi-hop adjacency matrixes $[\mathbf{I}, \mathbf{A}, \mathbf{A}^2, ..., \mathbf{A}^k]$ then regard $\mathbf{A}^{gpr}$ as the neighborhood and calculate the desired CM.

## E  Additional Detailed Implementation of CMGNN

**Overall Message Passing Mechanism.** The overall message passing mechanism in CMGNN is formatted as follows:

$$\mathbf{Z}^l = \text{diag}(\alpha_0^l)\mathbf{Z}^{l-1}\mathbf{W}_0^l + \text{diag}(\alpha_1^l)\hat{\mathbf{A}}\mathbf{Z}^{l-1}\mathbf{W}_1^l + \text{diag}(\alpha_2^l)(\mathbf{A}^{sup} \odot \mathbf{B}^{sup})\mathbf{Z}^{l-1}\mathbf{W}_2^l,$$
$$\mathbf{Z} = \mathop{\|}_{l=0}^{L} \mathbf{Z}^l, \tag{41}$$

where $\text{diag}(\alpha_0^l), \text{diag}(\alpha_1^l), \text{diag}(\alpha_2^l)\mathbb{R}^{N \times 1}$ are the learned combination weights introduced below.

**COMBNIE Function with Adaptive Weights.** Firstly, we list the aggregated messages $\widetilde{\mathbf{Z}}_r^l$ from 3 neighborhoods:

$$\widetilde{\mathbf{Z}}_0^l = \mathbf{Z}^{l-1}\mathbf{W}_0^l, \ \widetilde{\mathbf{Z}}_1^l = \hat{\mathbf{A}}\mathbf{Z}^{l-1}\mathbf{W}_1^l,$$
$$\widetilde{\mathbf{Z}}_2^l = (\mathbf{A}^{sup} \odot \mathbf{B}^{sup})\mathbf{Z}^{l-1}\mathbf{W}_2^l. \tag{42}$$

The combination weights are learned by an MLP with Softmax:

$$[\alpha_0^l, \alpha_1^l, \alpha_2^l] = \text{Softmax}(\text{Sigmoid}([\mathbf{Z}_0^l\|\mathbf{Z}_1^l\|\mathbf{Z}_2^l\|\mathbf{d}]\mathbf{W}_{att}^l)\mathbf{W}_{mix}^l), \tag{43}$$

where $\mathbf{W}_{att}^l \in \mathbb{R}^{(3d_r+1) \times 3}$ and $\mathbf{W}_{mix}^l \in \mathbb{R}^{3 \times 3}$ are two learnable weight matrixes, $\mathbf{d}$ is the node degrees which may be helpful to weights learning.

**The Message Passing of Supplementary Prototypes.** In practice, the virtual prototype nodes are viewed as additional nodes, which have the same message passing mechanism as real nodes:

$$\mathbf{Z}^{ptt,l} = \text{diag}(\alpha_0^{ptt,l})\mathbf{Z}^{ptt,l-1}\mathbf{W}_0^l + \text{diag}(\alpha_1^{ptt,l})\hat{\mathbf{A}}^{ptt}\mathbf{Z}^{ptt,l-1}\mathbf{W}_1^l$$
$$+ \text{diag}(\alpha_2^{ptt,l})(\mathbf{A}^{ptt,sup} \odot \mathbf{B}^{ptt,sup})\mathbf{Z}^{ptt,l-1}\mathbf{W}_2^l,$$
$$\mathbf{Z}^{ptt} = \mathop{\|}_{l=0}^{L} \mathbf{Z}^{ptt,l}, \tag{44}$$

where $\mathbf{A}^{sup,ptt} = \mathbf{1} \in \mathbb{R}^{K \times K}$ and $\mathbf{B}^{sup,ptt} = \hat{\mathbf{C}}^{ptt}\hat{\mathbf{M}}$ are similar with those of real nodes.

**Update Strategy for the Estimation of the Compatibility Matrix.** For the sake of efficiency, we do not estimate the compatibility matrix in each epoch. Instead, we save it as fixed parameters and only update it when the evaluation performance is improved during the training.

**Prediction of CMGNN.** CMGNN leverages the prediction of the model during message passing. For initialization, nodes have the same probabilities belonging to each class. During the message passing, the prediction soft label $\hat{\mathbf{C}}$ is replaced by the output of CMGNN, formatted as follow:

$$\hat{\mathbf{C}} = \text{CLA}((Z)), \tag{45}$$

where CLA is a classifier implemented by an MLP and $\mathbf{Z}$ is the final node representations.

## F  More Detail about the Benchmark

In this section, we describe the details of the new benchmarks, including (i) the reason why we need a new benchmark: drawbacks of existing datasets; (ii) detailed descriptions of new datasets; (iii) baseline methods and the codebase; and (iv) details of obtaining benchmark performance.

### F.1  Drawbacks in Existing Datasets

As mentioned in [30], the widely used datasets Cornell, Texas, and Wisconsin[2] have a too small scale for evaluation. Further, the original datasets Chameleon and Squirrel have an issue of data leakage,

---

[2]https://www.cs.cmu.edu/afs/cs.cmu.edu/project/theo-11/www/wwkb

where some nodes may occur simultaneously in both training and testing sets. Then, the splitting ratio of training, validation, and testing sets are different across various datasets, which is ignored in previous works.

Therefore, to build a comprehensive and fair benchmark for model effectiveness evaluation, we will newly organize 10 datasets with unified splitting across various homophily values in the next Subsection F.2.

## F.2 New Datasets

In our benchmark, we adopt ten different types of publicly available datasets with a unified splitting setting (48%/32%/20% for training/validation/testing) for fair model comparison, including **Roman-Empire** [30], **Amazon-Ratings** [30], **Chameleon-F** [30], **Squirrel-F** [30], **Actor** [15], **Flickr** [37], **BlogCatalog** [37], **Wikics** [38], **Pubmed** [39], and **Photo** [29]. The datasets have a variety of homophily values from low to high. The statistics and splitting of these datasets are shown in Table 6. The detailed description of the datasets is as follows:

Table 6: Statistics and splitting of the experimental benchmark datasets.

| Dataset | Nodes | Edges | Attributes | Classes | Avg. Degree | Undirected | Homophily | Train / Valid / Test |
|---|---|---|---|---|---|---|---|---|
| Roman-Empire | 22,662 | 65,854 | 300 | 18 | 2.9 | ✓ | 0.05 | 10,877 / 7,251 / 4,534 |
| Amazon-Ratings | 24,492 | 186,100 | 300 | 5 | 7.6 | ✓ | 0.38 | 11,756 / 7,837 / 4,899 |
| Chameleon-F | 890 | 13,584 | 2,325 | 5 | 15.3 | ✗ | 0.25 | 427 / 284 / 179 |
| Squirrel-F | 2,223 | 65,718 | 2,089 | 5 | 29.6 | ✗ | 0.22 | 1,067 / 711 / 445 |
| Actor | 7,600 | 30,019 | 932 | 5 | 3.9 | ✗ | 0.22 | 3,648 / 2,432 / 1,520 |
| Flickr | 7,575 | 479,476 | 12,047 | 9 | 63.3 | ✓ | 0.24 | 3,636 / 2,424 / 1,515 |
| BlogCatalog | 5,196 | 343,486 | 8,189 | 6 | 66.1 | ✓ | 0.40 | 2,494 / 1,662 / 1,040 |
| Wikics | 11,701 | 431,206 | 300 | 10 | 36.9 | ✓ | 0.65 | 5,616 / 3,744 / 2,341 |
| Pubmed | 19,717 | 88,651 | 500 | 3 | 4.5 | ✓ | 0.80 | 9,463 / 6,310 / 3,944 |
| Photo | 7,650 | 238,162 | 745 | 8 | 31.1 | ✓ | 0.83 | 3,672 / 2,448 / 1,530 |

- **Roman-Empire**[3] [30] is derived from the extensive article on the Roman Empire found on the English Wikipedia, chosen for its status as one of the most comprehensive entries on the platform. It contains 22,662 nodes and 65,854 edges between nodes. Each node represents an individual word from the text, with the total number of nodes mirroring the length of the article. An edge between two nodes is established under one of two conditions: the words are sequential in the text or they are linked in the sentence's dependency tree, indicating a grammatical relationship where one word is syntactically dependent on the other. Consequently, the graph is structured as a chain graph, enriched with additional edges that represent these syntactic dependencies. The graph encompasses a total of 18 distinct node classes, with each node being equipped with 300-dimensional attributes obtained by fastText word embeddings [40].

- **Amazon-Ratings**[3] [30] is sourced from the Amazon product co-purchasing network metadata dataset [41]. It contains 24,492 nodes and 186,100 edges between nodes. The nodes within this graph represent products, encompassing a variety of categories such as books, music CDs, DVDs, and VHS video tapes. An edge between nodes signifies that the respective products are often purchased together. The objection is to forecast the average rating assigned to a product by reviewers, with the ratings being categorized into five distinct classes. For the purpose of node feature representation, we have utilized the 300-dimensional mean values derived from fastText word embeddings [40], extracted from the textual descriptions of the products.

- **Chameleon-F** and **Squirrel-F**[3] [30] are specialized collections of Wikipedia page-to-page networks [42], of which the data leakage nodes are filtered out by [30]. Within these datasets, each node symbolizes a web page, and edges denote the mutual hyperlinks that connect them. The node features are derived from a selection of informative nouns extracted directly from Wikipedia articles. For the purpose of classification, nodes are categorized into five distinct groups based on the average monthly web traffic they receive. Specifically, Chameleon-F contains 890 nodes and 13,584 edges between nodes, with each node being equipped with 2,325-dimensional features. Squirrel-F contains 2,223 nodes and 65,718 edges between nodes, with each node being equipped with a 2,089-dimensional feature vector.

---

[3]https://github.com/yandex-research/heterophilous-graphs/tree/main/data

- **Actor**[4] [15] is an actor-centric induced subgraph derived from the broader film-director-actor-writer network, as originally presented by [43]. In this refined network, each node corresponds to an individual actor, and the edges signify the co-occurrence of these actors on the same Wikipedia page. The node features are identified through the presence of certain keywords found within the actors' Wikipedia entries. For the purpose of classification, the actors are organized into five distinct categories based on the words of the actor's Wikipedia. Statistically, it contains 7,600 nodes and 30,019 edges between nodes, with each node being equipped with a 932-dimensional feature vector.

- **Flickr** and **Blogcatalog**[5] [37] are two datasets of social networks, originating from the blog-sharing platform BlogCatalog and the photo-sharing platform Flickr, respectively. Within these datasets, nodes symbolize the individual users of the platforms, while links signify the followship relationships that exist between them. In the context of social networks, users frequently create personalized content, such as publishing blog posts or uploading and sharing photos with accompanying tag descriptions. These textual contents are consequently treated as attributes associated with each node. The classification objection is to predict the interest group of each user. Specifically, Flickr contains 7,575 nodes and 479,476 edges between nodes. The graph encompasses a total of 9 distinct node classes, with each node being equipped with a 12047-dimensional attribute vector. BlogCatalog contains 5,196 nodes and 343,486 edges between nodes. The graph encompasses a total of 6 distinct node classes, with each node being equipped with 8189-dimensional attributes.

- **Wikics**[6] [38] is a dataset curated from Wikipedia, specifically designed for benchmarking the performance of GNNs. It is meticulously constructed around 10 distinct categories that represent various branches of computer science, showcasing a high degree of connectivity. The node features are extracted from the text of the associated Wikipedia articles, leveraging the power of pretrained GloVe word embeddings [44]. These features are computed as the average of the word embeddings, yielding a comprehensive 300-dimensional representation for each node. The dataset encompasses a substantial network of 11,701 nodes interconnected by 431,206 edges.

- **Pubmed**[7] [39] is a classical citation network consisting of 19,717 scientific publications with 44,338 links between them. The text contents of each publication are treated as their node attributes, and thus each node is assigned a 500-dimensional attribute vector. The target is to predict which of the paper categories each node belongs to, with a total of 3 candidate classes.

- **Photo**[8] [29] is one of the Amazon subset network from [29]. Nodes in the graph represent goods and edges represent that two goods are frequently bought together. Given product reviews as bag-of-words node features, each node is assigned a 745-dimensional feature vector. The task is to map goods to their respective product category. It contains 7,650 nodes and 238,162 edges between nodes. The graph encompasses a total of 8 distinct product categories.

### F.3 Baseline Methods and the Codebase

For comprehensive comparisons, we choose 13 representative homophilous and heterophilous GNNs as baseline methods in the benchmark, including (i) Shallow base model: MLP; (ii) Homopihlous GNNs: GCN, GAT, GCNII; and (iii) Heterophilous GNNs: H2GCN, MixHop, GBK-GNN, GGCN, GloGNN, HOGGCN, GPR-GNN. Detailed descriptions of some of these methods can be seen in Appendix A.2.

To explore the performance of baseline methods on new datasets and facilitate future expansions, we collect the official/reproduced codes from GitHub and integrate them into a unified codebase. Specifically, all methods share the same data loaders and evaluation metrics. One can easily run different methods with only parameters changing within the codebase. The codebase is based on the PyTorch[9] framework, supporting DGL[10] and PyG[11]. Detailed usages of the codebase are available in the Readme file of the codebase.

---

[4] https://github.com/bingzhewei/geom-gcn/tree/master/new_data/film
[5] https://github.com/TrustAGI-Lab/CoLA/tree/main/raw_dataset
[6] https://github.com/pmernyei/wiki-cs-dataset
[7] https://linqs.soe.ucsc.edu/datac
[8] https://github.com/shchur/gnn-benchmark
[9] https://pytorch.org
[10] https://www.dgl.ai
[11] https://www.pyg.org

### F.4 Details of Obtaining Benchmark Performance

Following the settings in existing methods, we construct 10 random splits (48%/32%/20% for train/valid/test) for each dataset and report the average performance among 10 runs on them along with the standard deviation.

For all baseline methods except MLP, GCN, and GAT, we conduct parameter searches within the search space recommended by the original papers. The searches are based on the NNI framework with an anneal strategy. We use Adam as the optimizer for all methods. Each method has dozens of search trails according to their time costs and the best performances are reported. The currently known optimal parameters of each method are listed in the codebase. We run these experiments on NVIDIA GeForce RTX 3090 GPU with 24G memory. The out-of-memory error during model training is reported as OOM in Table 2.

## G More Details about Experiments

In this section, we describe the additional details of the experiments, including experimental settings and results.

### G.1 Additional Experimental Settings

Our method has the same experimental settings within the benchmark, including datasets, splits, evaluations, hardware, optimizer and so on as in Appendix F.4.

**Parameters Search Space.** We list the search space of parameters in Table 7, where patience is for early stopping, nhidden is the embedding dimension of hidden layers as well as the representation dimension $d_r$, relu_varient decides ReLU applying before message aggregation or not as in ACM-GCN, structure_info determines whether to use structure information as supplement node features or not.

Table 7: Parameters search space of our method.

| Parameters | Range |
|---|---|
| learning rate | {0.001, 0.005, 0.01, 0.05} |
| weight_decay | {0, 1e-7, 5e-7, 1e-6, 5e-6, 5e-5, 5e-4} |
| patience | {200, 400} |
| dropout | [0, 0.9] |
| $\lambda$ | {0, 0.01, 0.1, 1, 10} |
| layers | {1, 2, 4, 8} |
| nhidden | {32, 64, 128, 256} |
| relu_variant | {True, False} |
| structure_info | {True, False} |

**Ablation Study.** In the ablation study, there are three variants of our methods: without SM, without DL, without SM and DL. For "without SM", we delete the supplementary messages during message passing, using only messages from node ego and raw neighborhood for combination. For "without DL", we simply set $\lambda = 0$ to delete the discrimination loss. For "without SM and DL", we just combine the above two settings.

### G.2 Additional Experimental Results

In this subsection, we show some additional experimental results and analysis.

#### G.2.1 Additional Results of CM Estimations

The additional visualizations of CM estimations are shown in Figure 5. As we can see, our method can estimate quite accurate CMs among various homophily and class numbers, which provides a good foundation for the construction of supplementary messages.

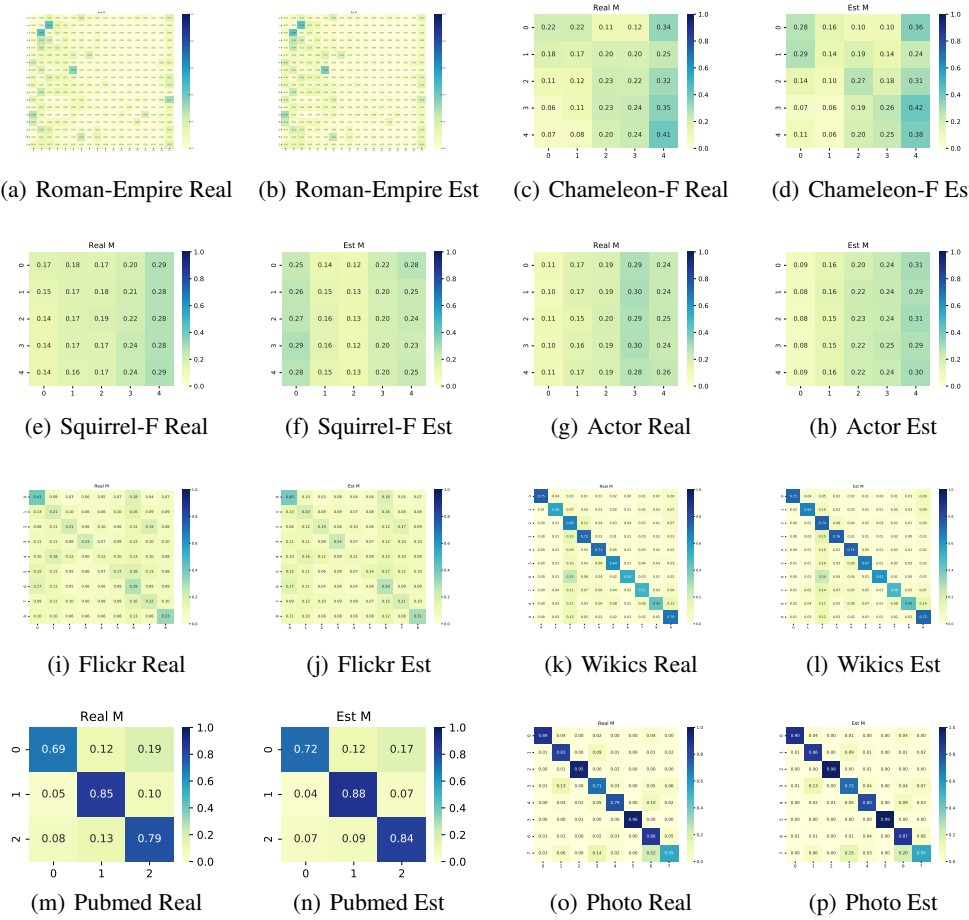

Figure 5: The visualization of real and estimated CMs on other datasets.

### G.2.2 Additional Performance on Nodes with Various Levels of Degrees.

We show the additional performance on nodes with various degrees in Table 8. The results show that CMGNN can achieve relatively good performance on low-degree nodes, especially on heterophilous graphs. For the opposite results on homophilous graphs, we guess it may be due to the low-degree nodes in homophilous graphs having a more discriminative semantic neighborhood, such as a one-hot form. On the contrary, there are relatively more high-degree nodes with confused neighborhoods due to the randomness, which leads to the shown results on homophilous graphs.

Table 8: Node classification accuracy comparison (%) among nodes with different degrees.

| Dataset | Roman-Empire | | | | | Chameleon-F | | | | | Actor | | | | |
|---|---|---|---|---|---|---|---|---|---|---|---|---|---|---|---|
| Deg. Prop.(%) | 0~20 | 20~40 | 40~60 | 60~80 | 80~100 | 0~20 | 20~40 | 40~60 | 60~80 | 80~100 | 0~20 | 20~40 | 40~60 | 60~80 | 80~100 |
| Ours | **88.60** | **87.00** | **85.59** | **86.25** | **74.33** | 40.73 | **45.28** | **56.02** | **46.64** | 39.93 | 35.56 | 37.14 | 38.40 | 36.03 | 36.84 |
| ACM-GCN | 79.00 | 77.87 | 73.52 | 72.09 | 53.77 | 39.51 | 41.21 | 52.25 | 45.80 | **47.09** | 34.48 | 36.58 | 36.27 | 34.63 | **37.46** |
| OrderedGNN | **88.60** | **87.00** | 85.56 | 84.68 | 69.69 | **43.21** | 44.51 | 49.16 | 38.27 | 32.23 | 35.94 | **38.06** | 37.87 | 35.77 | 37.15 |
| GCNII | 86.79 | 85.14 | 85.20 | 84.75 | 71.09 | 34.84 | 42.56 | 47.50 | 40.45 | 41.84 | **36.89** | 37.20 | 38.53 | **38.02** | 36.99 |
| Dataset | Squirrel | | | | | Pubmed | | | | | Photo | | | | |
| Deg. Prop.(%) | 0~20 | 20~40 | 40~60 | 60~80 | 80~100 | 0~20 | 20~40 | 40~60 | 60~80 | 80~100 | 0~20 | 20~40 | 40~60 | 60~80 | 80~100 |
| Ours | **45.37** | **47.10** | **45.25** | **34.86** | 37.10 | 89.32 | 89.33 | 89.31 | **92.62** | 89.39 | 88.88 | 95.76 | 96.96 | **98.27** | 97.55 |
| ACM-GCN | 41.12 | 44.30 | 44.22 | 32.97 | **42.10** | 89.60 | **89.54** | 89.58 | 92.02 | 89.23 | 89.88 | 95.20 | 96.95 | 98.00 | 97.56 |
| OrderedGNN | 43.78 | 45.53 | 43.09 | 27.90 | 28.48 | 89.67 | 89.37 | 89.45 | 92.54 | 89.02 | **90.13** | 95.77 | **97.14** | 98.24 | **97.58** |
| GCNII | 43.08 | 45.55 | 43.65 | 33.07 | 38.05 | **89.77** | 89.50 | 89.24 | 92.45 | 88.86 | 88.89 | 95.36 | 97.12 | 97.83 | 96.64 |

 **G.2.3 Efficiency Study**

**Complexity Analysis.** The number of learnable parameters in layer $l$ of CMGNN is $3d_r(d_r + 1) + 9$, compared to $d_r d_r$ in GCN and $3d_r(d_r + 1) + 9$ in ACM-GCN. The time complexity of layer $l$ is composed of 3 parts (i) AGGREGATE function: $O(Nd_r^2)$, $O(Nd_r^2 + Md_r)$ and $O(Nd_r^2 + NKd_r)$ for identity neighborhood, raw neighborhood and the supplementary neighborhood respectively, where $M = |\mathcal{E}|$ denotes the number of edges; (ii) COMBINE function: $O(3N(3d_r + 1) + 12N)$ for adaptive weights calculating and $O(3N)$ for combination; (iii) FUSE function: $O(1)$ for concatenations. To this end, the time complexity of CMGNN is $O(Nd_r(3d_r + K + 9) + Md_r + 18N + 1)$, or $O(Nd_r^2 + Md_r)$ for brevity.

**Experimental Running Time.** we report the actual average running time (ms per epoch) of baseline methods and CMGNN in Table 9 for comparison. The results demonstrate that CMGNN can balance both performance effectiveness and running efficiency.

Table 9: Effiency study results of average model running time (ms/epoch). OOM denotes out-of-memory error during the model training.

| Method | Roman-Empire | Amazon-Ratings | Chameleon-F | Squirrel-F | Actor | Flickr | BlogCatalog | Wikics | Pubmed | Photo |
|---|---|---|---|---|---|---|---|---|---|---|
| MLP | 7.8 | 7.0 | 6.1 | 6.5 | 6.3 | 9.1 | 6.7 | 6.4 | 6.1 | 5.8 |
| GCN | 33.8 | 33.4 | 7.9 | 20.6 | 34.4 | 37.2 | 30.4 | 25.5 | 35.6 | 28.1 |
| GAT | 15.9 | 67.3 | 10.3 | 14.0 | 30.8 | 66.2 | 17.6 | 26.8 | 33.4 | 36.0 |
| GCNII | 29.4 | 28.4 | 37.3 | 19.6 | 37.7 | 84.2 | 97.6 | 20.7 | 258.0 | 46.9 |
| H2GCN | 20.0 | 31.2 | 17.2 | 32.4 | 55.6 | 415.7 | 165.5 | 332.8 | 39.0 | 87.6 |
| MixHop | 434.6 | 486.3 | 21.9 | 31.0 | 30.6 | 90.4 | 81.6 | 277.4 | 89.5 | 172.2 |
| GBK-GNN | 119.8 | 191.8 | 31.0 | 238.1 | 157.9 | OOM | OOM | 198.6 | 137.0 | 193.3 |
| GGCN | OOM | OOM | 55.7 | 42.1 | 199.8 | 111.2 | 108.7 | 226.6 | 2290.8 | 105.2 |
| GloGNN | 25.4 | 19.3 | 121.8 | 23.3 | 1292 | 562.9 | 30.9 | 1658.1 | 43.2 | 677.4 |
| HOGGCN | OOM | OOM | 25.2 | 54.3 | 1002.9 | 707.3 | 367.4 | 1406 | OOM | 655.3 |
| GPR-GNN | 15.9 | 12.5 | 22.3 | 23.2 | 16.7 | 15.9 | 14.7 | 49.8 | 13.2 | 13.1 |
| ACM-GCN | 56.7 | 56.7 | 26.1 | 29.7 | 22.5 | 60.7 | 31.7 | 42.4 | 37.1 | 40.1 |
| OrderedGNN | 86.0 | 110.8 | 49.5 | 60.1 | 67.8 | 107.0 | 88.3 | 116.9 | 88.1 | 78.2 |
| **CMGNN** | 51.5 | 93.5 | 62.5 | 64.7 | 19.0 | 52.5 | 69.8 | 44.0 | 102.9 | 20.4 |

