# OpenReview forum: "Revisiting the Message Passing in Heterophilous Graph Neural Networks"
_NeurIPS.cc/2024/Conference — Submitted to NeurIPS 2024_

### Official Review · Reviewer_Fk2z · 2024-06-29

**Soundness:** 2
**Presentation:** 2
**Contribution:** 2
**Rating:** 5
**Confidence:** 3

**Summary:**

1. The paper unifies existing heterophilous graph neural networks (HTGNNs) into a Heterophilous Message-Passing (HTMP) mechanism.
2. The authors reveal that the effectiveness of HTMP is due to increasing differences among node representations belonging to different classes.
3. Guided by this revelation, the paper then introduces Compatibility Matrix-aware Graph Neural Network (CMGNN) to further enhance HTGNNs.
4. The authors conduct fair evaluations and comparative analysis on multiple benchmark datasets, highlighting the superior performance of the HTMP mechanism and the proposed CMGNN method.

**Strengths:**

1. The claims are supported empirically by a detailed comparison across multiple benchmark datasets.
2. The paper is well-written and clearly structured, with each section logically building on the previous ones.

**Weaknesses:**

1. Several research publications [1, 2, 3] have used compatible matrices to boost the effectiveness of GNNs on heterophilic graphs. In-depth qualitative and quantitative comparisons are missing from this submission. Including such analyses would significantly increase the importance of the contributions.
2. Some claims made in the paper, such as Observation 1 and Observation 2 in Section 4, would benefit from additional analysis. For instance, including theoretical analysis with formal notations would provide more rigorous support for these claims.
3. Existing survey articles have unified and categorised message passing on heterophilic graphs [4,5,6]. This submission should compare and position the proposed HTMP unification against these categorisations.
4. The experiments lack a comparison of training times with baseline models. Including an analysis of the tradeoff between accuracy and training time would greatly enhance the results.




References:
1. Simplifying Node Classification on Heterophilous Graphs with Compatible Label Propagation, In TMLR'22,
2. Explicit pairwise factorized graph neural network for semi-supervised node classification, In UAI'21,
3. Graph Neural Networks with Heterophily, In AAAI'21,
4. Graph Neural Networks for Graphs with Heterophily: A Survey,
5. Learning from Graphs with Heterophily: Progress and Future,
6. Heterophily and Graph Neural Networks: Past, Present and Future.



**Edit post Rebuttal:**
The authors have promised to include detailed comparisons in a future revised version. Since these details cannot be verified within the review period, I will lower my confidence from 4 to 3. However, given that other major concerns have been addressed, I will raise my rating by one point from 4 to 5.

**Questions:**

1. Were there insights on how the compatible matrices utilised in CMGNN compared with the methods in the referenced publications [1, 2, 3]? What were the pros and cons of CMGNN to tackle heterophily with existing methods that use compatibility matrices?
2. Were there theoretical foundations that support Observation 1 and Observation 2? What formal notations and theoretical analyses could be included to strengthen the support for these observations?
3. How did the proposed HTMP unification differ from the categorisations presented in the existing survey articles [4, 5, 6]? What unique insights did the HTMP unification offer compared to existing unifications?
4. What did the tradeoff between accuracy and training times of the proposed methods and the baselines look like?

**Limitations:**

The authors have provided some discussion on the limitations of their work (for instance, see section 7 on Page 9).

Potential negative societal impacts are not relevant to this study.

---

> ### Author Rebuttal · Authors · 2024-08-07
>
> Thanks for your insightful and constructive review of our work. The following are our detailed responses to the reviewer’s thoughtful comments. We are expecting these could be helpful in answering your questions.
>
> > Question #1 & Weakness #1: The connections and differences between CMGNN and existing methods that use compatibility matrices.
>
> Answer #1:
> Thanks for your suggestion! We would like to highlight that the usage of compatibility matrices in CMGNN **differs significantly from existing methods**.
>
> Existing methods [1, 2, 3] utilize the compatibility matrix (CM) to **redefine pair-wise relations (i.e. edge weights)** for existing edges, such as label propagation in CLP [1],  log-likelihood estimation in EPFGNN [2] and prior belief propagation in CPGNN [3]. In contrast, CMGNN leverages CM and virtual neighbors to **construct supplementary messages** while preserving the original neighborhood distribution.
> As a result, CMGNN benefits from its approach to utilizing CM in the following aspects compared with existing methods [1, 2, 3]:
> * **Better robustness for low-quality pseudo labels:** Existing methods utilize CM to guide the weights of propagation, which can lead to error accumulation with inaccurate pseudo labels. This is a common limitation of CM-based methods. In CMGNN, the CM is used to construct supplementary messages while original neighborhoods are preserved, mitigating the impact of inaccurate pseudo labels.
> * **Unlock the effectiveness of CM for low-degree nodes:** Existing methods redefine pair-wise relations only for existing edges, limiting the effectiveness of CMs for low-degree nodes. In CMGNN, virtual neighbors can provide prototype messages from every class, enhancing neighborhood messages for low-degree or even isolated nodes.
> * **More accurate estimation of CM:** While existing methods take naive approaches to estimate or initialize CM, CMGNN considers the effects of node degrees and model prediction confidence, resulting in more accurate CM estimation, especially in real-world situations. Additionally, CM in CMGNN is continuously updated with more accurate pseudo labels, creating a positive cycle.
>
> For quantitative comparison, we will add comparative experiments with these methods in the revised version, including performance, estimation accuracy of CM, efficiency, etc.
>
> > Question #2 & Weakness #2: Theoretical analysis for the observations in the paper.
>
> Answer #2:
> Thank you for your valuable comment! We have provided theoretical analyses supporting a core condition behind Observation 1 and 2: the discriminability of obtained representations is **positively correlated** with the discriminability among classes in CM. For detailed analyses, please refer to the global response due to space limits.
>
>
> > Question #3 & Weakness #3: The differences and unique insights of proposed HTMP compared to existing categorizations and unifications.
>
> Answer #3:
> Thanks for pointing out this! We delve into the message-passing mechanism in heterophilous GNN methods and propose a unification HTMP that **clarifies the intrinsic mechanisms** of these methods. In contrast, existing surveys [4, 5, 6] **provide a macroscopic view**, categorizing and unifying heterophilous GNNs with comprehensive but shallow analysis.
> Therefore, HTMP offers significant differences and unique insights compared to existing surveys [4, 5, 6]:
> * HTMP provides a **uniform symbolic form** and categorizes methods based on the values of component modules (e.g. neighborhood indicator and aggregation guidance). As a comparison, existing surveys categorize methods based on their ideas and designs, which are **described only by words** but are not limited to message-passing mechanisms (e.g. learning strategies).
> * As a result, the fine-grained HTMP can concretely **guide the design of new heterophilous message-passing mechanisms** through its modular architecture, whereas existing surveys **offer guidance primarily at the conceptual level**.
>
> The detailed differences and connections with each work [4,5,6] are as follows:
> * **Survey [4]:** This work categorizes the designs of heterophilous GNNs into non-local neighbor extensions and GNN architecture refinement. The first group corresponds to part of the neighborhood indicator in HTMP, while the second group includes some designs of HTMP functions (AGGREGATE, COMBINE and FUSE). It organizes heterophilous designs directly, whereas HTMP provides systematic categorizations from a message-passing perspective.
> * **Survey [5]:** This work focuses on learning from heterophilous graphs, where message passing is only a minor aspect of its taxonomy. Thus, it offers a broader view, whereas HTMP is more specialized in message-passing mechanisms.
> * **Survey [6]:** This survey examines the impact of heterophilous graph characteristics on GNNs. For categorizations, it simply lists some effective designs in heterophilous GNNs, whereas HTMP offers a clearer categorization of these and additional designs.
>
>
>  > Question #4 & Weakness #4: The tradeoff between accuracy and training times of CMGNN and baselines.
>
> Answer #4:
> Thank you for this suggestion! We have followed your suggestion and visualized the tradeoff between accuracy and training time in the supplemental **PDF of the global response**. From the results in the figures, we can find that our proposed **CMGNN achieves the best performance while maintaining relatively low computational complexity**. Compared to the top-2 performed baselines (OrderedGNN, GCNII), CMGNN has superior classification performance and lower time consumption.

---

> > ### Comment · Reviewer_Fk2z · 2024-08-12
> > **Thanks for the Rebuttal**
> >
> > Thanks to the authors for the rebuttal.
> >
> > The responses mostly address the concerns raised. The paper should include in-depth qualitative and quantitative comparisons with existing methods that use compatibility matrices. This should involve performance metrics, robustness tests, and case studies to highlight the differences and advantages of the proposed method compared to similar methods in the literature.
> >
> > The authors have promised to include these comparisons in a future revised version. Since these details cannot be verified within the review period, I will lower my confidence. However, given that all other concerns have been addressed, I will raise my rating by one point.

---

> ### Author Response · Authors · 2024-08-13
> **Thank you**
>
> Thank you for engaging in the discussion.
>
> We greatly appreciate your constructive comments, which have contributed to the improvement and solidity of our work!
>
> Best regards!

---

### Official Review · Reviewer_GM4Z · 2024-07-08

**Soundness:** 3
**Presentation:** 2
**Contribution:** 3
**Rating:** 5
**Confidence:** 3

**Summary:**

This work revisits the message-passing mechanisms in existing HTGNNs and reformulates them into a unified heterophilous message-passing (HTMP) mechanism. Based on HTMP, the authors propose a new framework named CMGNN. Experiments on 10 datasets with 13 different baseline models demonstrate the effectiveness of the proposed framework.

**Strengths:**

1. This work proposes a unified heterophilous message-passing (HTMP) mechanism, which could be a guideline for further research on heterophilous GNN.
2. Based on the HTMP mechanism, this work proposes a new framework named CMGNN, which is novel and has basic value.
3. The effectiveness of the HTMP mechanism and CMGNN framework is well supported by experiment results.

**Weaknesses:**

1. Paper presentation could be further improved. For example, the conception of "good" heterophily and "bad" homophily deserves further explanation. There are some spelling mistakes, such as "heterophilious" in line 12.
2. If space permits, I feel like moving experiments in Appendix C to the main body would be better for the introduction of *Observation 1*.
3. It might be hard to follow as this paper has so many equations, especially those about CMGNN. So I suggest providing a flow chart or a pseudo algorithm for better understanding.
4. Conclusions in this paper are mainly based on experiment results. It would be better if corresponding theoretical analysis or proofs are provided.

**Questions:**

1. What are the contributions to the newly built benchmark datasets? It seems that this work just collects 10 existing datasets and sets them with the same train-valid-test split ratio.
2. The weighting function (Eq.9) seems to be quite empirical. Is it possible to further discuss it?

**Limitations:**

As the authors mentioned, this work mainly focuses on semi-supervised settings, which could be further generalized.

---

> ### Author Rebuttal · Authors · 2024-08-07
>
> We sincerely appreciate your thoughtful review of our paper. Below, we address the reviewer's concerns point by point, hoping that a better understanding of every point can be delivered.
>
> > Question #1: The contributions of benchmark datasets and unified codebase.
>
> Answer #1:
> To address the issues of method comparison caused by drawbacks like data leakages and extremely imbalanced classes in existing datasets, we have collected and **filtered suitable graph datasets** from heterophilous GNNs methods and other fields (e.g. Anomaly Detection). This collection spans various levels of homophily, **providing a robust foundation for performance evaluation**.
> However, the benchmark datasets are not the main contribution of the paper. Instead, we consider them an additional resource to assist the community in better evaluating methods.
>
> In addition to addressing dataset limitations, we have built a unified heterophilous GNN codebase, which has the following contributions:
> * We have gathered the official and reproduced codes of 13 representative baseline methods and integrated them along with our CMGNN into a unified PyTorch-based codebase. All methods share the same call interfaces, ensuring a fair comparison environment.
> * The codebase will be open-sourced, **enabling easier research and further development of this field**, such as quickly evaluating baseline methods on new datasets.
>
> > Question #2: The explanation and discussion of Eq.9.
>
> Answer #2:
> Eq.9 defines a weighting function considering the effects of node degrees. The core idea is that **nodes with lower degrees correspond to lower weights** during compatibility matrix estimation, as high-degree nodes usually have representative neighborhoods while low-degree nodes often have incomplete ones. However, the relationship between weights and node degrees should not be linear. For low-degree nodes, increases in degree should yield more significant benefits compared to high-degree nodes. Beyond a certain threshold, increases in degree yield tiny benefits.
>
> We have empirically chosen $K$ and $3K$ as fixed thresholds for the weighting function to simplify the design **without multiple attempts**. This approach is straightforward and can be substituted with other forms that meet the same criteria. It allows for further design but is not a priority compared to other modules.
>
>
> > Weakness #1: Suggestions for paper presentation.
>
> Answer #3:
> Thank you for these suggestions. We will work on improving the paper presentation and correcting the spelling errors in the revised version. Here is a detailed explanation of "good" heterophily and "bad" homophily:
> * The conception of "good" heterophily, introduced by [1], is based on empirical observations and highlights the existence of different kinds of heterophily. Specifically, **GCNs can achieve strong performance in "good" heterophily settings**, where the compatibility matrix exhibits strong discriminability.  This concept is qualitative rather than quantitative.
> * "Bad" homophily describes a scenario where, despite having more neighbors from the same class than from any other classes, the **homophily level is insufficient** for vanilla message-passing methods (GCN, GAT) to outperform MLPs.
>
> Reference:
>
> [1] Is Homophily a Necessity for Graph Neural Networks? in ICLR 2022.
>
>
> > Weakness #2: Suggestion for paper architecture.
>
> Answer #4: Thanks for this suggestion. We will consider it in the revised version.
>
> > Weakness #3: The flow chart and algorithm of CMGNN.
>
> Answer #5: Thanks for your suggestion. We have followed your suggestions and included the figure and algorithm of CMGNN in the **PDF of the global response**, which will be added in the revised version to better illustrate the architecture of CMGNN.
>
>
>
> > Weakness #4: Theoretical analysis for the observations in the paper.
>
> Answer #6:
> Thanks for your suggestion! We have provided theoretical analyses and support for a core condition underlying Observation 1 and 2: the discriminability of obtained representations is **positively correlated** with the discriminability among classes in CM.
> For detailed analyses, please refer to the **global response** due to the space limits.

---

### Official Review · Reviewer_cjsi · 2024-07-13

**Soundness:** 3
**Presentation:** 3
**Contribution:** 3
**Rating:** 6
**Confidence:** 5

**Summary:**

This paper aims to address the question of "why does message passing remain effective on heterophilous graphs" and proposes a unified framework called heterophilous message-passing (HTMP) mechanism. It extensively reviews the architecture of existing heterophilous GNNs under this framework. It then moves on to discuss the empirical observation that the success of message passing in existing heterophilous GNNs is attributed to their implicitly enhancement of the compatibility matrix among classes, and proposed a new GNN approach called CMGNN to further enhance the separability of the compatibility matrix for different classes in the message passing process. The paper includes an extensive empirical analysis involving 10 benchmark datasets and 13 well-established baseline GNNs, and show that the proposed CMGNN approach has the best overall performance against the baselines.

**Strengths:**

- The writing is clear and well-organized for most parts of the paper;
- The paper gives an extensive survey of existing message-passing GNNs under the HTMP mechanism in Table 1 and Appendix A.
- The experiments are well-thought and extensive: it addresses the drawbacks of the previous homophilous and heterophilous node classification benchmarks identified in previous works by using more recent benchmark datasets, and include 13 baselines for a comprehensive evaluation of the proposed method.
- The proposed approach, CMGNN, has the best overall performance against 13 baselines on 10 benchmark datasets.

**Weaknesses:**

- This work builds upon the findings of several previous works regarding the effective designs for GNNs under heterophily and when is heterophily challenging (or in other words, "bad") for GNNs. While the authors cited these works in some parts of the paper ([6,9,12,18] in References), I feel that **some of the observations in the paper overlapped with the findings in previous works, and their connections and differences are not clearly stated in the paper**.
  - For example, Observation 1 seems to overlap with the previous observations made in [6] ("to ensure that the neighborhood patterns for nodes with different labels are distinguishable, the inter- class similarity should be low") and [9] ("two key factors, low-degree nodes and complex compatibility matrices, deteriorate the distinguishability of the neighborhood label distributions when coupled with heterophily, thus making heterophily a unique challenge for GNNs in most cases").
  - Given this, I also think that the claim in the related work section (line 732-734) that "these reviews ... not exploring the reason behind the effectiveness of message passing in heterophilous graphs" is inaccurate, as this paper is in fact built upon these analyses regarding the effectiveness of message passing in heterophilous graphs.

- Section 5 (method) is too condensed to present a clear picture of how the proposed Compatibility Matrix-Aware GNN (CMGNN) works for the readers. For example, it is unclear what "topology structure" that the authors are considering as "additional available node features", and the term in Eq. 7 is not well explained. The authors also didn't explain clearly in the main paper how is the "soft pseudo labels" being generated for the model. It will help with the understanding if the authors can include a figure showing the architecture of the proposed CMGNN model. I feel the "method" section is the most novel part in the paper and deserves more length in the paper.

- It would be good to analyze the computational complexity and/or compare the empirical runtime of the model with the baselines.

- As a minor point, the "Norm" term in Eq. 3 should be explained as "L1 normalization for matrix row vectors" to avoid the confusion that the normalization is done with the L1 norm for *matrix* (instead of for vectors).

**Questions:**

- As per Weakness point 1, can you describe how the observations in the paper are connected to, and different with, prior observations in [6,9,12,18]?
- As per Weakness point 2, what is the "topology structure" that you considered as "additional available node features"? How does the use of these features affect the performance of the proposed model?
- Can you provide some analysis regarding the computational complexity and/or the empirical runtime of the model?

**Limitations:**

The authors acknowledged the limitation that the proposed HTMP framework is only applicable to GNNs following the message-passing mechanism. One additional limitation is that the paper is mostly empirical and does not give theoretical underpinnings.

---

> ### Author Rebuttal · Authors · 2024-08-07
>
> We would like to thank you for your deeply thorough review. We have carefully considered your comments and suggestions, and the following are our detailed responses.
>
> > Question #1 & Weakness #1: The connections and differences between the observations in this paper and prior works.
>
> Answer #1:
> Thanks for pointing out the omission. **Our principal finding is Obs.2 with Obs.1 serving as a foundational context to enhance the understanding of Obs.2.** The detailed comparisons are as follows:
> * Prior works [6, 9] have primarily analyzed the challenging issue of heterophily in vanilla GNNs. **While their conclusions are similar to our Obs.1, as they emphasize data, our focus is more on the message passing.** However, they did not explore the reasons behind the effectiveness of heterophilous message passing (HTMP). Further, prior works [12, 18] investigate the effectiveness of specific designs separately. In contrast, our Obs.2 encompasses a comprehensive set of various effective designs in HTMP and attributes their effectiveness to a unified goal. Consequently, **our Obs.2 provides a more comprehensive and unified perspective** for understanding the working mechanism of HTMP.
> * Upon the above analysis, we also recognize that there are inaccuracies in our related works section that could lead to misunderstandings. A more accurate description would be: "These reviews do not explore the unified reasons for the effectiveness of various designs in heterophilous message passing." We will correct this in the revised version.
>
>
> > Question #2 & Weakness #2: More detailed description of the method and "topology structure".
>
> Answer #2:
> Thank you for the suggestions! We completely agree that a figure can significantly aid readers in understanding the whole method. **Following your suggestions, we have included a figure in the PDF of the global response.**
> The detailed descriptions of "topology structure" are as follows:
> * **The explanation of "topology structure":** The term "topology structure" refers to the connection relationship among nodes, represented by the adjacency matrix $\mathbf{A}$. Each row $\mathbf{A}_i$ can be viewed as an additional $N$-dimensional feature for the corresponding node $i$. The inclusion of additional features is optional, depending on the observed performance on the validation set, as it may introduce extra computational cost and potentially redundant information.
> * **The explanation of terms in Eq.7:** In Eq.7, $\mathbf{Z}^0$ is the input for the first layer of message passing and can be obtained in two ways: (1) by using the topology structure as additional features, with $\mathbf{W}^{X}\in \mathbb{R}^{d_f \times d_r}$, $\mathbf{W}^{A} \in \mathbb{R}^{N \times d_r}$ and $\mathbf{W}^{0} \in \mathbb{R}^{2d_r \times d_r}$ as learnable weight matrices; (2) by using only attribute features, where $\mathbf{W}^{0} \in \mathbb{R}^{d_f \times d_r}$ is a learnable weight matrix.
> * **The effect of additional structural features:** The additional structural features offer another way to utilize connection relationships, introducing both discriminant and redundant information. Thus **it presents a trade-off between the advantages and disadvantages**. We conducted an ablation study to examine its effects and report the results in the table below. The additional structure features have positive effects on five datasets while others are negative. It doesn’t significantly impact performance except for Roman-Empire. Moreover, **CMGNN can still achieve competitive results without using additional structural features**.
>
> |structural_features|Roman-Empire|Amazon-Ratings|Chameleon-F|Squirrel-F|Actor|Flickr| BlogCatalog|Wikics|Pubmed|Photo|
> |:---:|:---:|:---:|:---:|:---:|:---:|:---:|:---:|:---:|:---:|:---:|
> |True|68.43±2.23|**52.13±0.55**|**45.70±4.92**|**41.89±2.34**|35.72±0.75|**92.66±0.46**|96.47±0.58|**84.50±0.73**|88.90±0.45|95.08±0.43|
> |False|**84.35±1.27**|51.41±0.57|44.85±5.64|40.49±1.55|**36.82±0.78**|92.05±0.75|**97.00±0.52**|83.88±0.75|**89.99±0.32**|**95.48±0.29**|
>
>
>
> > Question #3 & Weakness #3: Computational complexity analysis and empirical runtime comparison.
>
> Answer #3:
> Thanks for your suggestion! The analysis of computational complexity and empirical runtime comparison are described as follows:
> * **Computational complexity analysis:** The computational complexity of layer $l$ consists of 3 parts: (i) AGGREGATE function: $O(N{d_r}^2)$, $O(N{d_r}^2+Md_r)$ and $O(N{d_r}^2+NKd_r)$ for identity, raw and the supplementary neighborhood, respectively, where $N$ and $M=|\mathcal{E}|$ denote the number of nodes and edges, $d_r$ is the dimension of representations; (ii) COMBINE function: $O(3N(3d_r+1)+12N)$ for calculating adaptive weights and $O(3N)$ for combination; (iii) FUSE function: $O(1)$ for concatenations. Thus, the time complexity of $L$-layer CMGNN is $O(L(Nd_r (3d_r+K+9)+Md_r+18N)+1)$, or $O(LN{d_r}^2+LM d_r)$ for brevity.
> * **Empirical runtime comparison:** Following your suggestions, we have visualized the tradeoff between accuracy and empirical runtime compared to baseline methods in the PDF of the global response. The results show that **CMGNN achieves the best performances with relatively low time consumption**. Compared with OrderedGNN and GCNII, which have the second- and third-best average ranks, CMGNN offers both better accuracy and lower time consumption.
>
>
> > Weakness # 4: Correction of the "Norm" term.
>
> Answer #4: Thank you for pointing out this mistake. We will correct it in the revised version.

---

> > ### Comment · Reviewer_cjsi · 2024-08-13
> > **Response to Authors' Rebuttal**
> >
> > I appreciate the authors detailed response to my questions and comments. After reading authors' response, I decide to keep my original rating.

---

> > > ### Author Response · Authors · 2024-08-13
> > >
> > > Thanks again for your careful reading and valuable suggestions, which have significantly improved our manuscript.
> > >
> > > Best Regards!

---

### Author Rebuttal · Authors · 2024-08-07

We sincerely thank all reviewers for their insightful and valuable review points. We add the figure and algorithm of the proposed CMGNN and the results of empirical runtime comparisons in the **PDF file** attached to this global response. We appreciate that some reviewers suggested we provide theoretical analyses of the observations and we provide detailed analyses as below. Due to limited space, some experimental results and answers to the reviewers' questions are in individual replies.


### Theoretical analyses of observation 1 and 2

Behind **Observation 1** and **2**, there is a core condition to support the conclusions:

> **Condition 1. The discriminability of obtained representations is **positively correlated** with the discriminability among classes in CM.**

Based on **Condition 1**, vanilla message passing (VMP) can work well with discriminative CM regardless of homophily levels and heterophilous message passing (HTMP) can achieve better performance by enhancing the discriminability of CM.
To prove this condition, we start with an assumption:

> **Assumption 1. The semantic neighborhood $C^{nb}$ of each node follows a class-specific distribution guided by CM, where** $C_{i} ^{nb}= \frac{\sum_{j\in \mathcal{N}(i)} C_{j}}{|\mathcal{N}(i)|}$ **indicates the proportion of neighbors from each class in node i's neighborhood.**

According to **Assumption 1**, the discriminability in CM is positively correlated with the discriminability in semantic neighborhoods.
Thus, if the message-passing mechanism can preserve the discriminability of the semantic neighborhood in the obtained representations, then **Condition 1** holds.
It would be sufficient if each distinct semantic neighborhood corresponds to a different output representation, in other words, the message-passing mechanism is an injective function for modeling semantic neighborhoods.

We further state an assumption:

> **Assumption 2. The input node messages $\mathbf{Z}^{l-1}$ of the message-passing layer exhibit clustering characteristics, where the average distance within a class is significantly smaller than the average distance between different classes. Meanwhile, the clustering centers of each class's input messages exist, formatted as $K$ prototypes $\\{ \mathbf{c}_k| k\in 1,...,K \\}$.**

This implies that the input messages of nodes from the same class are linearly correlated within a certain range of error.
Taking the most general mean aggregation as an example, we have the following theorem:

> **Theorem 1. Let $\text{MEAN}(\{\mathbf{Z}^{l-1}_j|j\in\mathcal{N}(i)\})$ be mean operator that aggregates neighbor messages for node $i$. Function $\text{MEAN}(\{\mathbf{Z}^{l-1}_j|j\in\mathcal{N}(i)\})$ is approximately injective if it is satisfied that all class prototypes $\mathbf{c}_k$ are orthogonal to each other.**

The injectivity ensures that each element in the domain of the input (i.e. semantic neighborhoods and neighbor messages) has a distinct and unique output in the output domain.
We find that as long as the conditions of **Theorem 1** are satisfied, the mean aggregation can be regarded as an injective function within a certain range of error.
Thus, the whole message-passing mechanism can be an approximately injective function for modeling the semantic neighborhoods when the COMBINE function is also injective, which can be easily satisfied.

In practice, the orthogonality of prototypes is hard to be satisfied completely but the difference among prototypes is still significant.
Thus, even if the message-passing mechanism is not completely injective, most of the discriminability can be preserved, making **Condition 1** hold.



#### **Proof of Theorem 1**.
We have the following lemma:

> **Lemma 1:. Injectivity is equivalent to null space equals $\{0\}$. Let $T\in \mathcal{L}(V,W)$, $T$ is injective if and only if $null(T)=\{0\}$.**

##### **Proof of Lemma 1**.

**Sufficiency**:
First, suppose $T$ is injective. We want to prove that $null(T)=\{0\}$.
We already know that $\{0\}\subset null(T)$.
Suppose $v\in null(T)$, then $T(v)=0=T(0)$.
Because $T$ is injective, the equation implies that $v=0$.
Thus we can conclude that $null(T)=\{0\}$, as desired.

**Necessity**:
Suppose $null(T)=\{0\}$, $u,v \in V$.
If $T(u)=T(v)$, then $T(u)-T(v)=T(u-v)=0$.
Thus $u-v=0$, which implies that $u=v$.
Hence $T$ is injective, as desired.

Having **Lemma 1** proved, now we express the mean aggregation in the form of $\mathbf{PZ}^{nb}=\mathbf{b}$, where $\mathbf{P}\in \mathbb{R}^{1\times |\mathcal{N}(i)|}$ denotes the mean aggregation operator, $\mathbf{Z}^{nb}\in \mathbb{R}^{|\mathcal{N}(i)| \times d_r}$ is the matrix consist of neighbor messages and $\mathbf{b}$ is the resulting representation.
Assuming that the messages of neighbors from the same class are linearly dependent, we can rewrite the equation as $\mathbf{P}'\mathbf{Z}^{p}\approx b$, where $\mathbf{P}'\in\mathbb{R}^{1\times K}$ is a weighted mean operator, $\mathbf{Z}^p\in\mathbb{R}^{K\times d_r}$ is a matrix consisting of the prototypes $\\{\mathbf{c}_k|k\in 1,...,K\\}$ of $K$ classes.
The injectivity of mean aggregation operator $\mathbf{P}$ involves considering the solution for $\mathbf{P}'\mathbf{Z}^{p}=0$.
Clearly, if it is satisfied that all $\mathbf{c}_k$ are orthogonal to each other, the null space $null(\mathbf{P}')=\{0\}$, indicating that the mean aggregation operator is approximately injective, as desired in **Theorem 1**.


The above analyses provide theoretical support for **Observation 1** and **2**, proving the feasibility of HTMP to improve performance by enhancing the discriminability of CM.


Finally, we once again thank all reviewers for their insightful comments which are very helpful for improving the quality of our paper.
All discussions, supplementary experiments and figures will be included in our revised version. If any remaining questions have not been resolved, please feel free to continue the discussion with us!

---

### Decision · Program_Chairs · 2024-09-25

**Decision:**

Reject

**Comment:**

This paper revisits ideas for addressing heterophily in graph neural networks and proposes a "unified heterophilous message-passing (HTMP) mechanism". Overall concerns about heterophily in graphs addressed by GNNs is a rough area of research with confusion about relevant notions of heterophily, whether heterophily is itself a primary problem or merely a distraction from other relevant aspects of graph structure that differentiate datasets of interest. The reviewers appreciated the paper's detailed experiments but raised a number of concerns, including the lack of clairty about "good" vs "bad" heterophily, claims that require greater theoretical backing, insufficient contexutalization of the paper in previous work on heterophily, and missing comparisons with relevant baselines. After a discussion and the raising of a reviewer score, the paper remains on the low side of borderline.